# The membrane curvature-inducing REEP1-4 proteins generate an ER-derived vesicular compartment

Yoko Shibata [1,2] ✉, Emily E. Mazur[1,2], Buyan Pan [1,2], Joao A. Paulo [1], Steven P. Gygi [1], Suyog Chavan[1], L. Sebastian Alexis Valerio [1], Jiuchun Zhang[1] & Tom A. Rapoport [1,2] ✉

The endoplasmic reticulum (ER) is shaped by abundant membrane curvature-generating proteins that include the REEP family member REEP5. The REEP1 subfamily, consisting of four proteins in mammals (REEP1-4), is less abundant and lack a N-terminal region. Mutations in REEP1 and REEP2 cause Hereditary Spastic Paraplegia, but the function of these four REEP proteins remains enigmatic. Here we show that REEP1-4 reside in a unique vesicular compartment and identify features that determine their localization. Mutations in REEP1-4 that compromise curvature generation, including those causing disease, relocalize the proteins to the bulk ER. These mutants interact with wild-type proteins to retain them in the ER, consistent with their autosomal-dominant disease inheritance. REEP1 vesicles contain the membrane fusogen atlastin-1, but not general ER proteins. We propose that REEP1-4 generate these vesicles themselves by budding from the ER, and that they cycle back to the ER by atlastin-mediated fusion. The vesicles may serve to regulate ER tubule dynamics.

The endoplasmic reticulum (ER) is an essential organelle in all cells, consisting of a dynamic, continuous membrane network of tubules and planar sheets[1–5]. The relative abundance of tubules and sheets varies among cell types and even within a single cell. This is exemplified in neurons, whose dendrites and cell bodies contain a mixed morphology of tubules and sheets, while their axons are filled solely with tubules of narrow diameter[6]. The high membrane curvature of tubule cross-sections and sheet edges is stabilized by two distinct, conserved protein families, the Reticulons and a branch of the REEPs (receptor expression-enhancing proteins) that in mammals includes REEP5 and the less characterized REEP6[7–9]. The Reticulons and REEP5 have redundant functions in shaping the ER. They are abundant in all eukaryotic cells and share similar features, possessing four transmembrane segments (TMs) followed by an amphipathic helix near the C-terminus (APH-C). Both features are required to generate high membrane curvature[10–13]. The shaping and dynamics of the ER also

require conserved membrane fusion GTPases. In mammals, they are called atlastins and have several isoforms (ATL1-3). These membrane-bound GTPases mediate the tethering and fusion of ER tubules to form the three-way junctions of the polygonal network[14–19].

The REEP family contains another branch which in mammals consists of four proteins, REEP1-4[20]. REEP1-4 proteins are highly conserved among eukaryotes and generate high membrane curvature in vitro[21], but their cellular function remains enigmatic. They have been implicated in ER morphology[20], but they likely play a limited role in tubule formation, as they are much less abundant than the Reticulons and REEP5[22]. They have also been linked to lipid droplet formation[23,24], mitotic ER dynamics[25], nuclear pore formation[26], and most recently, to macroautophagy in fission yeast[21,27,28]. Mammalian REEP1-4 share high sequence similarity amongst each other but differ from the REEP5 subfamily in their structures, as they lack REEP5's N-terminal region, including the first TM, and possess a disordered C-terminal

[1]Department of Cell Biology, Harvard Medical School, 240 Longwood Avenue, Boston, MA 2115, USA. [2]Howard Hughes Medical Institute, Boston, MA, USA. ✉e-mail: yoko_shibata@hms.harvard.edu; tom_rapoport@hms.harvard.edu

tail[21]. The core 'REEP' domains of the two subfamilies share high homology: REEP1's three TMs are predicted to have a similar structure as the last three TMs of REEP5 and to be followed by an APH-C (Fig. 1a). Like REEP5, REEP1-4 proteins homodimerize through the TM domain, and both the TMs and APH-C are required for membrane curvature generation[21].

REEP1-4 proteins are ubiquitously expressed in mammalian cells, although different members are expressed in distinct cell types[22,29]. REEP1 and REEP2 are enriched in neurons, and mutations in these proteins are linked to hereditary spastic paraplegia (HSP) and distal hereditary motor neuropathy type Vb (HMN5B), two related, inherited neuropathies caused by axonal shortening of motor neurons[30–36]. HSP

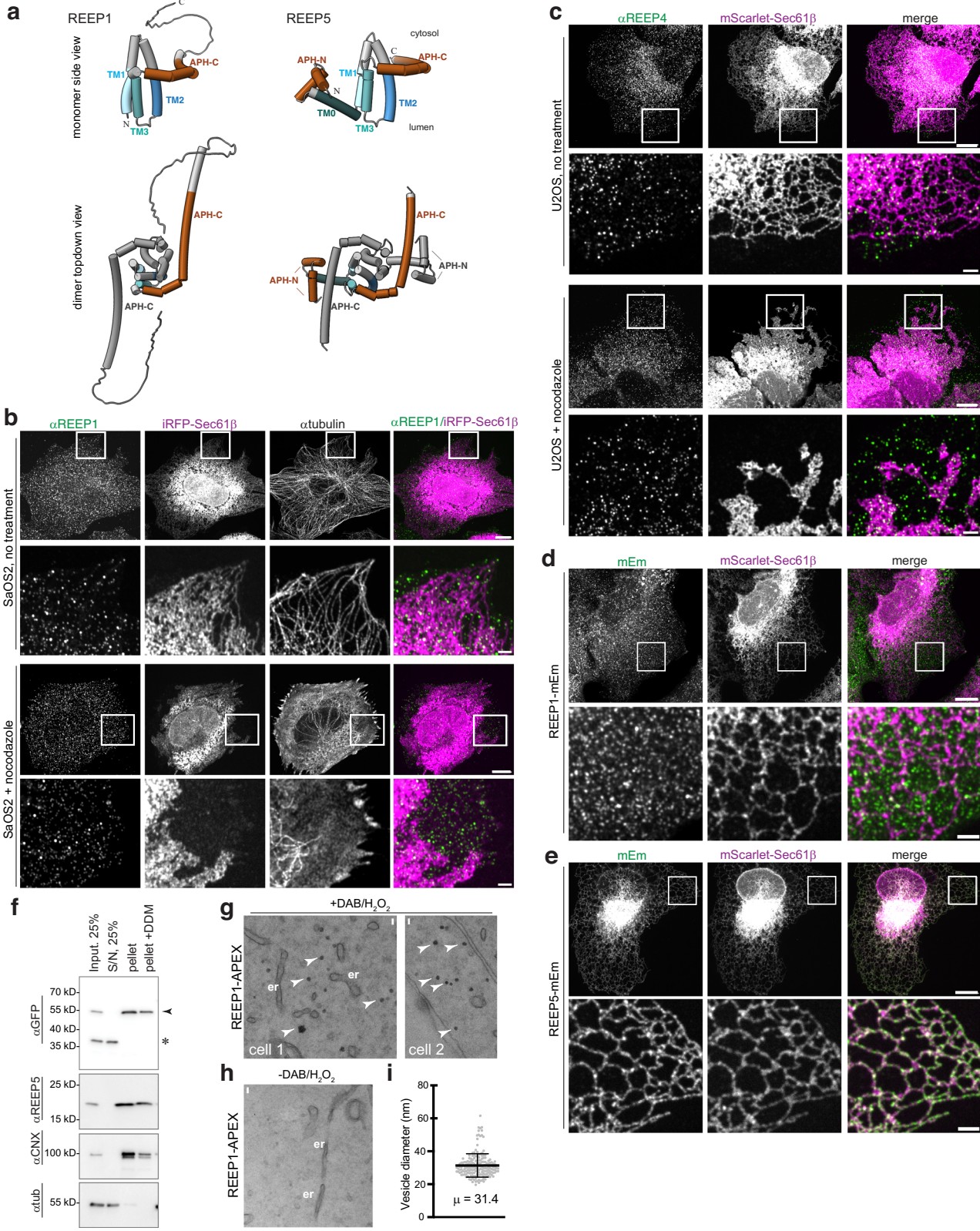

**Fig. 1 | REEP1 and REEP4 localize to vesicles in mammalian cells. a** Alphafold-predicted models of the monomer and homodimer of human REEP1 and REEP5. Transmembrane (TM) segments are numbered and colored in blue. Amphipathic helices at the N- or C-termini (APH-N or -C, respectively) are in sienna. The second monomer in the dimer is shown in gray. **b** Endogenous REEP1 localization in SaOS2 cells stably expressing the ER marker Sec61β fused to near-infrared fluorescent protein (iRFP-Sec61β). Cells were analyzed by indirect immunofluorescence using anti-REEP1 (αREEP1) and anti-alpha tubulin (αtubulin) antibodies, and imaged using confocal fluorescence microscopy. Shown are whole cells and enlargements of the boxed region. Right column shows overlays of αREEP1 (green) and iRFP-Sec61β (magenta). Bottom two rows show cells after 50 μM nocodazole treatment. **c** Endogenous REEP4 localization was determined in U2OS cells stably expressing mScarlet-Sec61β (magenta) by immunostaining with anti-REEP4 antibodies (αREEP4, green). Right panels show overlays, and bottom two rows show cells after nocodazole treatment. **d** U2OS cells stably co-expressing REEP1-mEmerald (REEP1-mEm, green) and the ER marker mScarlet-Sec61β (magenta) were imaged using confocal fluorescence microscopy. Right panels show overlays. Bottom panels show magnifications of the boxed region. **e** As in (**d**), but with cells stably expressing REEP5-mEmerald (REEP5-mEm) and transiently transfected with mScarlet-Sec61β. **f** Membrane fractionation of U2OS cells stably expressing REEP1-mEm. Cell lysates were fractionated into supernatant (S/N) and membrane pellet by ultra-centrifugation, and the pellet was solubilized in 1% dodecylmaltoside (DDM) detergent. Samples were analyzed by SDS-PAGE and immunoblotting with antibodies against mEmerald (αGFP), REEP5 (αREEP5), the membrane protein calnexin (αCNX), and cytosolic alpha tubulin (αtub). Arrowhead denotes full-length REEP1-mEm; asterisk, mEm fragment. **g** U2OS cells stably expressing REEP1 fused to ascorbate peroxidase (REEP1-APEX) were fixed, treated with diaminobenzidine and hydrogen peroxide (+DAB/$H_2O_2$), and analyzed by thin section electron microscopy. Arrowheads point to electron-dense, circular structures in +DAB/$H_2O_2$ samples. er, endoplasmic reticulum. Scale bars, 50 nm. **h** As in (**g**), but without DAB/$H_2O_2$ treatment (-DAB/$H_2O_2$). **i,** Quantification of the diameters of REEP1-APEX positive structures observed in (**g**). Shown are the mean and standard deviation. *n*, 186. For all scale bars (**b**–**e**), whole cell, 10 μm; magnification, 2 μm.

is also linked to mutations in ATL1[37], consistent with the observation that ATLs physically interact with REEP proteins[14,20]. Given the physiological significance of this branch of the REEP family, it is important to identify the function of these proteins.

Here, we show that a large fraction of REEP1-4 molecules resides at steady state in a unique vesicular compartment and not the bulk ER, as hitherto assumed. The distinct localization is caused by the lack of the N-terminal domain found in the REEP5 subfamily. The vesicles appear derived from the ER and are likely generated by REEP1-4 themselves through their membrane curvature-inducing activity, rather than by conventional vesicle budding in the secretory pathway. ATLs form a physical complex with REEP1 subfamily members and allow the vesicles to fuse back to the ER. REEP1 mutations in the APH-C that compromise curvature generation, including those causing disease, lead to the retention of REEP1 in the bulk ER. They act dominantly by interacting with wild-type REEP1-4 proteins to retain them in the ER, thus inhibiting vesicle formation. We hypothesize that REEP1-4 vesicles regulate the dynamics of the tubular ER network.

## Results

### REEP1-4 proteins localize to vesicles

We first used specific antibodies to localize endogenous REEP1 in SaOS2 and Bewo cells, two cell lines that express REEP1 at high levels[29]; for comparison, fluorescent protein fusions of Sec61β were used as ER markers[10] (Fig. 1b; Supplementary Fig. 1a, b). In both cell types, REEP1 localized to numerous punctae dispersed throughout the cytoplasm. No staining was seen after depletion of REEP1 with RNA interference (RNAi, Supplementary Fig. 1a), demonstrating the specificity of the antibody. In contrast to the punctate staining of REEP1, endogenous REEP5 showed the expected tubular ER staining (Supplementary Fig. 1b). In areas where the reticular network was well resolved, these punctae were often in close proximity to the ER; however, quantification in SaOS2 cells showed that at least 40% of them showed no ER overlap (Supplementary Fig. 1c). Because of resolution limits of immunofluorescence and light microscopy, this number likely underestimates the fraction of REEP1 that is distinct from the ER. Similar results were obtained for endogenous REEP2 in SKN-SH cells (Supplementary Fig. 1d) and endogenous REEP4 in U2OS cells (Fig. 1c; Supplementary Fig. 1e, note that no staining is detected in *REEP4* knock-out cells). Like REEP1, a substantial pool of REEP4 punctae (~25%) did not overlap with the ER network (Supplementary Fig. 1c). These results suggest that the REEP1-4 proteins do not exclusively localize to the bulk ER, as previously assumed[20,25]. This conclusion is supported by experiments in which we treated cells with nocodazole to depolymerize microtubules, which leads to a collapse and retraction of the ER network[10,38]: Many of the punctae remained at the cell

periphery while the bulk ER retracted towards the cell center (Fig. 1b, c, bottom two rows).

The punctate localization of REEP1 was also observed in U2OS cells stably expressing a mEmerald-tagged version of REEP1 (REEP1-mEm). REEP1-mEm was found in numerous punctae distinct from the bulk ER (labeled with a mScarlet fusion of Sec61β, mScarlet-Sec61β) (Fig. 1d; Supplementary Fig. 2a, b). In contrast, REEP5-mEm localized to the tubular ER and was depleted in ER sheets and the nuclear envelope (Fig. 1e), consistent with its preference for high membrane curvature regions[7,8,10]. The REEP1 punctae correspond to a membrane compartment, as REEP1-mEm sedimented with membranes and could be solubilized from membranes with a mild detergent (Fig. 1f).

To quantify the punctate localization of REEP1 with various ER markers, we calculated Pearson's correlation coefficients. As expected, comparing different ER markers with each other or with REEP5 showed strong positive correlation (Pearson's *r*-values > 0.5, Supplementary Fig. 2c). However, the localization of REEP1-mEm correlated only weakly with that of the ER proteins mScarlet-Sec61β, endogenous calnexin (a general ER membrane protein[9,39]; αCNX), endogenous Reticulon-4 (a tubular ER membrane protein[9]; αRTN4), endogenous luminal proteins[9] (αKDEL), or stably expressed luminal RFP-KDEL[39] (Supplementary Fig. 2c). Similarly, stably expressed REEP1 tagged with mCherry (REEP1-mCh) localized to punctae (Supplementary Fig. 2d) that did not significantly correlate with endogenous membrane or luminal ER markers (Supplementary Fig. 2c). As with the endogenous proteins, REEP1-mEm and -mCh punctae remained in the cell periphery after nocodazole treatment, while the ER retracted to the cell center (Supplementary Fig. 2e–h). The REEP1 punctae were also distinct from ER membranes stained with the lipophilic dye DiOC6 (ref. 1), regardless of whether the microtubules were intact or depolymerized (Supplementary Fig. 2g, h), indicating that they are not connected by a lipid bilayer with the ER. Cells stably expressing REEP1-mEm or -Ch grew normally (Supplementary Fig. 3a), did not display signs of apoptosis or ER stress (Supplementary Fig. 3b–i), and had normal, intact ER morphology (Fig. 1d; Supplementary Fig. 2a, b, g), confirming that the punctate REEP1 structures are not generated by aberrant ER fragmentation in unhealthy cells.

The same punctate pattern was also observed when untagged REEP1 was overexpressed by transient transfection in U2OS cells (Supplementary Fig. 4a). Pearson's correlation analysis confirmed that REEP1 maintained ER-independent punctate localization over a wide range of expression levels (Supplementary Fig. 4b). Overexpression of untagged REEP2 or REEP4 in U2OS gave similar results (Supplementary Fig. 4c, d), and a punctate pattern with limited ER overlap was also seen when hemagglutinin (HA)-tagged REEP1-4 were transiently expressed (Supplementary Fig. 4e). While it was previously reported that

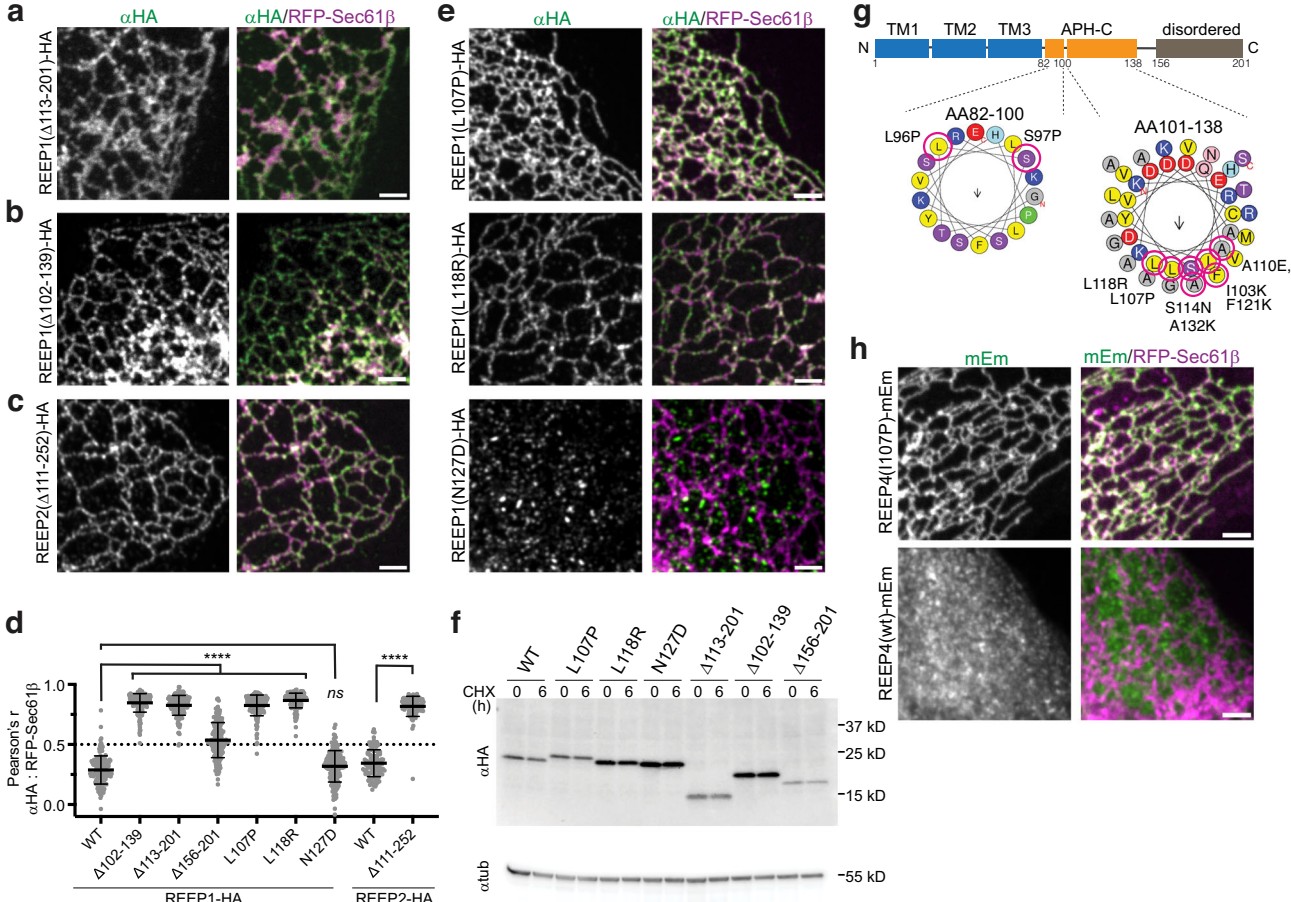

**Fig. 2 | Vesicle localization of REEP1-4 proteins is dependent on their APH-C.** **a** U2OS cells stably expressing the ER marker Sec61β fused to red fluorescent protein (RFP-Sec61β) were transfected with hemagglutinin (HA)-tagged REEP1 carrying the disease mutation Δ113-201, immunostained with anti-HA (αHA) antibodies, and imaged by confocal fluorescence microscopy. Right panel shows an overlay between αHA (green) and RFP-Sec61β (magenta). Scale bar, 2 μm. **b** As in (**a**), but with cells transfected with the REEP1 disease mutant Δ102-139. **c** As in (**a**), but with cells transfected with the REEP2 disease-linked mutant Δ111-252. **d** Pearson's correlation coefficients ($r$) measuring colocalization of RFP-Sec61β with wild-type or APH-C disease mutants of HA-tagged REEP1 or REEP2. Shown are means and standard deviations. $n$, REEP1wt, 153; REEP1Δ102-139, 164; REEP1Δ113-201, 166; REEP1Δ150-201,166; REEP1L107P, 168; REEP1L118R, 171; REEP1N127D, 163; REEP2WT, 106; REEP2 Δ111-252, 102 cells. $P$-values for REEP1 were calculated using one-way ANOVA analysis, multiple comparisons (Dunnett's method). REEP2 $p$-values were calculated using an unpaired two-tailed $t$ test. ****$p < 0.0001$; ns, not significant. Exact $p$-values are listed in the Source Data file. **e** As in (**a**), but in cells transfected with the REEP1-HA constructs carrying the L107P disease mutation, or putative disease mutations L118R or N127D. **f** The stabilities of C-terminal REEP1-HA mutants were analyzed by cycloheximide (CHX) chase. Cells transfected with wild-type or mutant HA-tagged REEP1 were collected at timepoint 0 or after CHX treatment for 6 h, and lysates were analyzed by immunoblotting with αHA and anti-tubulin (αtub) antibodies. **g** Schematic of REEP1 domain organization and APH-C prediction. C-terminal REEP1 point mutations that cause relocalization to the bulk ER are circled in red. Amphipathic helices were predicted using heliquest (https://heliquest.ipmc.cnrs.fr/)[70]. The arrows indicate the hydrophobic moments. Note that all mutations are either helix breakers or reside along the APH-C's hydrophobic face. **h** *REEP4* knockout U2OS cells stably expressing mEm-fused REEP4 (I107P) (top panels) or wild-type REEP4 (bottom panels) and transfected with RFP-Sec61β were imaged by confocal microscopy. Right panels show overlays between mEm (green) and RFP-Sec61β (magenta). Scale bars, 2 μm.

overexpressed REEP1 predominantly localizes to bundled ER tubules[20], we observed bundled tubules only in cells with the highest levels of expression; moreover, the tubules did not colocalize with RFP-Sec61β (Supplementary Fig. 4f, g). The distinct punctate localization of REEP1-4 proteins was also observed in Hela cells (Supplementary Fig. 4h). Notably, when REEP1 or REEP4 was overexpressed in U2OS cells, the number of punctae increased concomitantly with expression levels (Supplementary Fig. 4i–k), suggesting that these proteins generate the punctate structures themselves.

To further characterize these punctae, REEP1 was stably expressed as a fusion with ascorbate peroxidase 2 (REEP1-APEX) in U2OS cells, and the protein's localization was examined by thin-section electron microscopy after treatment with diaminobenzidine and $H_2O_2$, which forms an electron-dense precipitate[40]. REEP1-APEX localized to circular vesicle-like structures that have an average diameter of approximately 30 nm (Fig. 1g–i). Live-cell imaging of REEP1-mEm or REEP1-mCh in

U2OS cells showed that the majority of punctae move rapidly and independently of the tubular ER, although a few punctae appeared attached to the ER for several minutes (Supplementary Videos 1-4). Taken together, these data indicate that a substantial pool of REEP1-4 molecules localizes to a vesicular compartment that is distinct from the bulk ER.

## Vesicular localization of REEP1-4 proteins depends on the membrane curvature-generating APH-C

We next tested whether the APH-C affects the cellular localization of REEP1, as this region is essential for the generation of high membrane curvature[21] and many mutations linked to HSP or HMN5B map to it[34–36] (Supplementary Table 1). We found that the disease-causing APH-C deletion mutants, Δ113-201 and Δ102-139, did not localize to vesicles but rather to the bulk ER, where they overlapped with RFP-Sec61β (Δ113-201 and Δ102-139 vs. wt; Fig. 2a, b vs. Supplementary Fig. 4e).

Similarly, a disease-linked APH-C deletion mutant of REEP2 relocalized from vesicles to the ER (Δ111-252 vs. wt; Fig. 2c vs. Supplementary Fig. 4e). These colocalizations were confirmed by quantifying Pearson's correlation coefficients (Fig. 2d).

We next screened through disease-linked point mutations in the APH-C (see Supplementary Table 1 for provenance). The L107P mutant, which has been established to cause HSP, localized to the bulk ER rather than to vesicles (Fig. 2e, quantification in Fig. 2d). Several other mutants that are less clinically established also localized to the ER network (L118R; Fig. 2e, quantification in Fig. 2d; L96P, S97P, A110E, S114N, Supplementary Fig. 5a), while mutants with changes in the hydrophilic side of the APH or in the disordered tail had no effect (N127D, Fig. 2e, quantification in Fig. 2d; R124Q, G125S, T131A, T131I, V134M, G142R, R147I, P173L, K181T, Supplementary Fig. 5b). Cycloheximide-chase experiments showed that APH mutations causing relocalization of REEP1-HA to the ER do not overtly destabilize the proteins (Fig. 2f), making it unlikely that they are retained in the ER due to misfolding. All point mutations that caused relocalization of REEP1-HA from vesicles to the ER are alpha helix breakers or introduce hydrophilic or charged residues into the hydrophobic face of the APH-C (depicted in Fig. 2g). A similar ER relocalization was also seen when an APH mutation was introduced into REEP4-mEm (I107P, analogous to L107P of REEP1, Fig. 2h).

To analyze more systematically the features required in the APH-C for REEP1 vesicular localization, we introduced other deletions (Supplementary Fig. 5c) or mutated residues along the APH-C's hydrophobic face into lysines (Supplementary Fig. 5d). All changes caused the relocalization of REEP1-HA to the bulk ER except for the most distal mutation, A132K, which had an intermediate phenotype (Supplementary Fig. 5d). The deletion of the disordered C-terminal tail (Δ156-201) also caused a mild redistribution to the ER (Supplementary Fig. 5c, quantification in Fig. 2d).

Taken together, our results show that the APH-C of REEP1-4 proteins is required for vesicular localization and that this localization is physiologically important given the linkage to disease. It is likely that the hydrophobic side of the APH-C dips into the cytosolic leaflet of the lipid bilayer, displacing phospholipid molecules and thereby generating high membrane curvature. APH-C mutations would compromise lipid insertion, leading to defects in curvature-generation.

## The TM domain is important for REEP1 vesicle localization
Disease-causing mutations are also found in the membrane-spanning region of REEP1, where the majority map to the highly conserved cytosolic loop between TM1 and TM2[30–32,35] (Supplementary Fig. 6a). As reported previously[41], some of these mutations caused the relocalization of REEP1-HA to lipid droplets (P19R, A20E, W42R), although several also led to redistribution to the bulk ER (P19L, S23F, T55K; Supplementary Fig. 6b–d). One mutation, D56N, had no effect on localization. All TM domain mutations that cause REEP1 relocalization to the bulk ER/lipid droplets were unstable, as shown by cycloheximide-chase experiments (Supplementary Fig. 6e). Thus, as proposed previously[32], these mutations likely cause disease through loss of function. None of the mutants, except D56N, colocalized with vesicles marked by REEP1-mEm stably expressed in the same cells (Supplementary Fig. 6f–m; quantification in Supplementary Fig. 6n), suggesting that they do not form dimers with the wild-type protein. The mutated residues map to predicted interaction regions within and between the REEP1 monomers (Supplementary Fig. 6o) and are likely essential for intramolecular folding and homodimerization.

## Features that cause the distinct localization of REEP1 and REEP5
Next, we asked what structural features cause the distinct localization of REEP1 and REEP5 to vesicles and the tubular ER, respectively. REEP5 has a unique N-terminal domain with two predicted APHs and an additional TM segment (Fig. 1a, Fig. 3a). Deleting this domain (Δ2-51)

relocalized REEP5 to punctae that poorly overlapped with the ER marker RFP-Sec61β (wt vs. Δ2-51, Fig. 3b, c; quantification in Fig. 3d). Interestingly, these REEP5 Δ2-51 punctae did not colocalize with those marked by REEP1 (Fig. 3e), suggesting that the REEP5 mutant and REEP1 generate their own distinct vesicles. As in the case of REEP1, the vesicular localization of the REEP5 Δ2-51 mutant depended on the APH-C: A double mutant with an inactive APH-C (Δ2-51; V156P) was enriched in the ER (Fig. 3f, quantification in Fig. 3d). When the N-terminal domain of REEP5, including its two APHs and the first TM, was fused to full-length REEP1, the chimera also predominantly localized to the ER (Fig. 3g, quantification, Fig. 3d). Taken together, these results indicate that the unique N-terminus of REEP5 is a major determinant for ER tubule localization and is responsible for the distinct localizations of REEP1 and REEP5.

## REEP1 APH mutants can relocalize wild-type REEP1-4 from vesicles to the ER
Because REEP1-4 proteins form homodimers through their TM domains[21], we wondered if the ER-localized REEP1 APH-C mutants could retain wild-type REEP1 in the ER. We tested this possibility by co-expressing wild-type or mutant REEP1-HA with wild-type REEP1-mEm, using tandem constructs in which the two coding sequences are separated by the ribosomal skip site P2A (Fig. 4a). As expected, wild-type REEP1-HA colocalized with REEP1-mEm in ER-independent punctae and the TM domain mutant P19R had no effect on wild-type REEP1-mEm localization (Fig. 4b, c). However, the ER-localized APH-C mutants of REEP1-HA (L107P or Δ102-139) redistributed REEP1wt-mEm to the bulk ER, although some ER-independent vesicles remained (Fig. 4d, e). Pearson's correlation quantification of these colocalizations confirmed these conclusions (Fig. 4f).

Given that the TM domain is highly conserved among REEP1-4, we next tested whether the disease mutant REEP1 L107P could also retain other REEP1 paralogs to the ER. To this end, we generated a stable cell line that expresses a mCherry fusion of REEP1 L107P (REEP1L107P-mCh) and analyzed the localization of transfected HA-tagged wild-type REEP1-4 in these cells. REEP5-HA was used as a control. Consistent with our tandem construct analyses (Fig. 4d, f), both REEP1L107P-mCh and wild-type REEP1-HA localized to the bulk ER, where they overlapped with endogenous calnexin (Fig. 4g, Pearson's correlation quantifications in Fig. 4h–j). Similarly, in the presence of REEP1L107P-mCh, HA-tagged wild-type REEP2-4 also localized to the ER (Fig. 4g; quantifications in Fig. 4h–j), and all REEP1-4 proteins were as enriched in the ER as REEP5-HA (Fig. 4i, j). These data indicate that the REEP1 L107P mutant can interact with all members of the REEP1-4 subfamily, retaining them in the bulk ER and thereby acting in a dominant-negative manner. The ER retention of the wild-type proteins also suggests that the REEP1-containing vesicles originate from the ER.

Consistent with heterodimerization, all REEP1-4 family members resided in the same ER-independent vesicles, as HA-tagged wild-type REEP1-4 colocalized with stably expressed wild-type REEP1-mCh in punctae (Fig. 4h–j). In contrast, REEP5-HA maintained its localization to the tubular ER and did not affect the punctate localization of REEP1-mCh (Fig. 4h–j), suggesting that REEP5 does not interact with the REEP1-4 subfamily.

## REEP1 vesicles comprise a unique vesicular compartment
We next compared the localization of the REEP1 vesicles to that of known cellular structures. While the vesicles often aligned along microtubules, they showed little association with F-actin (Supplementary Figs. 2e, 7a). We found no major colocalization of REEP1 with vesicles corresponding to early, late, or recycling endosomes (GFP-Rab5, -Rab7, or -Rab11, respectively), or to vesicles marked by the autophagy protein ATG9A[42] (Fig. 5a–d; quantification in Fig. 5e). Additionally, no extensive colocalization was seen with mitochondria (αcyto c or Tom20) or Golgi (αgiantin) (Supplementary Figs. 3c; 7b, c);

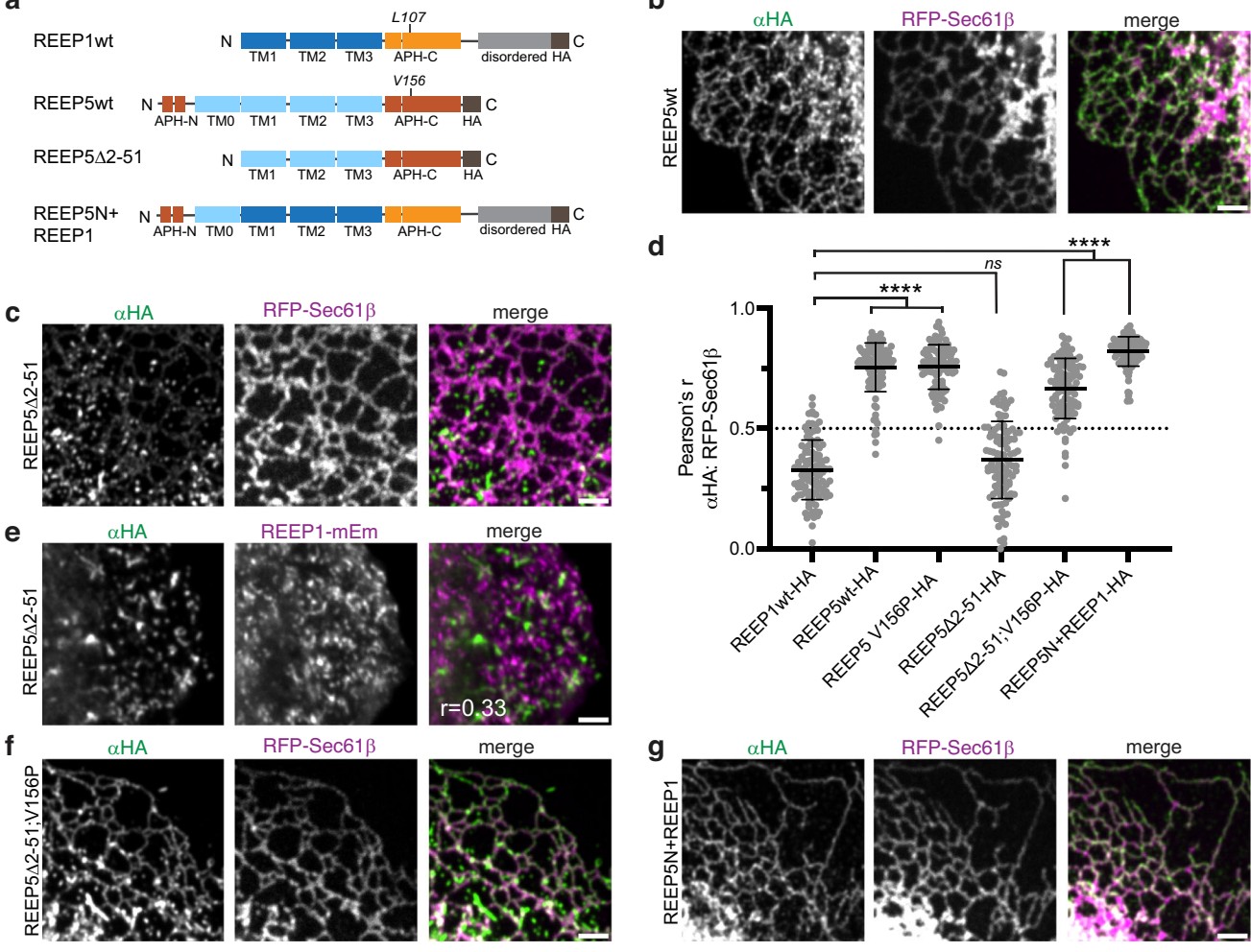

**Fig. 3 | REEP5's unique N-terminal domain is important for tubular ER localization. a** Schematics of REEP5 N-terminal deletion and chimera constructs used in (**b**–**g**). Wild-type (wt) REEP1 is shown for comparison. Note that residue V156 in REEP5 corresponds to L107 in REEP1. **b** U2OS cells stably expressing RFP-Sec61β were transfected with HA-tagged wild-type REEP5 (REEP5wt) and analyzed by immunostaining with αHA antibodies. Right panel shows the overlay between αHA (green) and RFP-Sec61β (magenta). Scale bar, 2 μm. **c** As in (**b**), but with transfection of a REEP5-HA construct lacking N-terminal residues 2-51 (REEP5Δ2-51). **d** Colocalization quantifications of HA-tagged wild-type REEP1, REEP5, or REEP5 mutants with RFP-Sec61β in samples as analyzed in (**b**, **c**, **f**, **g**). REEP5-HA carrying the V156P mutation (homologous to REEP1L107P) was also analyzed. Pearson's correlation coefficients were measured between αHA and RFP. Shown are means

and standard deviations. n, REEP1wt-HA, 102; REEP5-HA, 111; REEP5V156P-HA, 101; REEP5Δ2-51-HA, 108; REEP5Δ2-51;V156P-HA, 102; REEP5N + REEP1-HA, 107 cells. *P*-values were calculated using one-way ANOVA analysis, multiple comparisons (Dunnett's method). ****$p < 0.0001$; ns, not significant. Exact p-values are listed in the Source Data file. **e** As in (**c**), but with cells stably expressing REEP1-mEm (magenta in the overlay). The Pearson's r-value between αHA and mEm localizations is shown in the inset. **f** As in (**b**), but with transfection of a REEP5Δ2-51 construct additionally carrying the V156P mutation in the APH-C (REEP5Δ2-51; V156P). **g** As in (**b**), but with transfection of a chimera construct consisting of the first 51 N-terminal residues of REEP5 fused to the N-terminus of wild-type REEP1 (REEP5N + REEP1).

with PIPEROsomes[43] (Fig. 5f); or with lipid droplets (Lipidtox), phosphatidylinositol synthase (mCh-PIS)[43], ER exit sites (αSec31), or peroxisomes (αPMP70, Pex3-mCh) (Fig. 5g–j, quantification in Fig. 5e). In some cells, REEP1 vesicles were often adjacent to peroxisomal membranes marked with Pex3-mCh (Supplementary Fig. 7d), but the reasons for the partial overlap are unclear. Additionally, REEP1 vesicle localization remained unaltered when ER-to-Golgi transport was inhibited with Brefeldin A[44] or by incubation at 16 °C[45] (Supplementary Fig. 7e, f), suggesting that they are not formed by conventional vesicular budding along the secretory pathway.

Given that the REEP1 ortholog Rop1 plays a role in macroautophagy in fission yeast, we also tested whether starvation affects REEP1 localization in U2OS cells. REEP1-mEm and -mCh remained distinct from the bulk ER, other vesicular compartments, or lipid droplets under starvation conditions (Supplementary Fig. 2c; Fig. 5e). However, a small fraction of stably expressed REEP1-mEm was found in

starvation-induced phagophore membranes marked by the autophagy proteins LC3 and DFCP1[46] but lacking the lysosomal protein LAMP1 (Supplementary Fig. 7g), and a minor pool of fluorescently-tagged REEP1 also colocalized with early phagophore membranes marked by both ATG9A and LC3 or Atg16 and LC3 (Supplementary Fig. 7h, i). However, the majority of REEP1-mEm or -mCh vesicles remained distinct from these markers, as well as from other membrane proteins involved in autophagy (VMP1[47], TMEM41[48], Stx17[49]), whether or not cells were starved (Supplementary Fig. 7g, h; Fig. 5d, e).

Finally, to directly determine the protein composition of these vesicles, we used isobaric labeling and quantitative proteomics. Human 293 cells were generated expressing either wild-type REEP1 or the ER-localized L107P mutant as a control, tagged with mEm-3xHA (REEP1wt-mEm-HA or REEP1L107P-mEm-HA). The wild-type and mutant constructs were genomically integrated into a single recombination site to achieve equal expression levels using a Flp

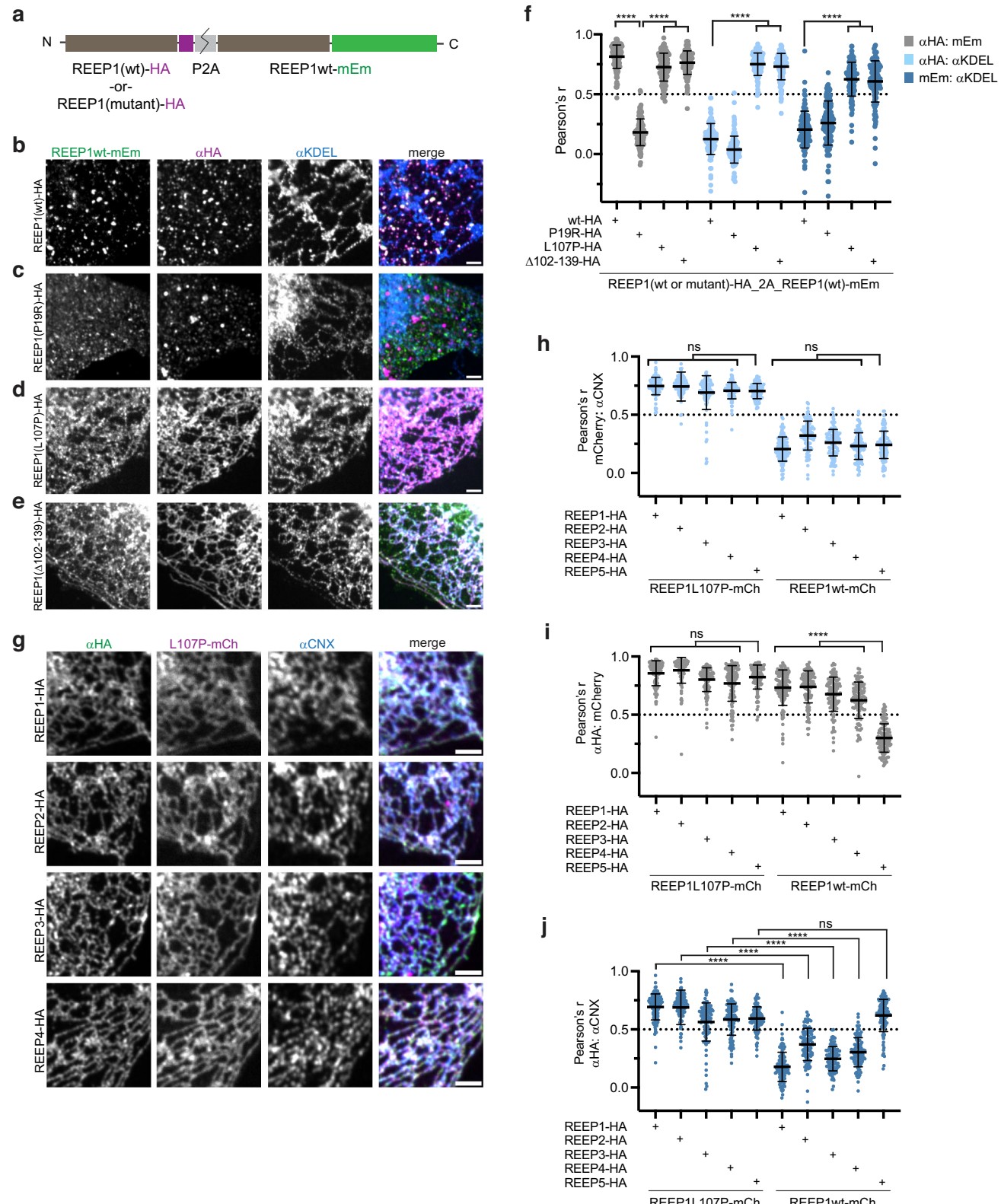

recombinase system (Flp-In), and homogenates from these cell lines were subjected to immunoprecipitation with anti-HA antibodies in the absence of detergent. The immuno-isolated membranes did not contain markers of lysosomes (LAMP1), endosomes (Rab5 and Rab11), ATG9A vesicles, or COPII (Sec31), as shown by immunoblotting (Supplementary Fig. 8a). Interestingly, however, even wild-type REEP1-mEm-HA precipitated membranes containing tubular ER markers

(REEP5 and ATL3), suggesting that a pool of REEP1 is closely associated with the ER and may even cycle through it (see below). Consistent with this idea, analysis of the samples by tandem mass tag-labeling and mass spectrometry showed that many ER proteins were only modestly depleted from membranes containing wild-type REEP1 (Supplementary Fig. 8b; Supplementary Data 1). A small number of proteins were enriched in REEP1wt membranes: Among the enriched proteins were

**Fig. 4 | REEP1 APH-C mutants can relocalize wild-type REEP1-4 to the bulk ER.** **a** Schematic of tandem REEP1 constructs analyzed in (**b**–**f**). Constructs encode for HA-tagged wild-type (wt) or mutant REEP1, the ribosomal skip site P2A, and wt REEP1-mEm. **b** U2OS cells were transfected with the REEP1wt-HA_P2A_REEP1wt-mEm construct, immunostained with αHA and αKDEL antibodies, and imaged using confocal microscopy. Right panel shows the overlay of mEm (green), αHA (magenta), and αKDEL (blue). **c** As in (**b**), but with the REEP1(P19R)-HA tandem construct. **d** As in (**b**), but with the REEP1(L107P)-HA tandem construct. **e** As in (**b**), but with the REEP1(Δ102-139)-HA tandem construct. **f** Colocalization quantification of tandem REEP1 constructs, as in (**b**–**e**). Pearson's correlation coefficients between αKDEL with mEm, αHA with αKDEL, and mEm with αKDEL in the same cells were quantified. Shown are means and standard deviations. $n$, REEP1wt-HA, 104; REEP1P19R-HA, 119; REEP1L107P-HA, 119; REEP1Δ102-139-HA, 128 cells. $P$-values were calculated using one-way ANOVA analysis, multiple comparisons (Dunnett's method). ****$p < 0.0001$. **g** U2OS cells stably expressing REEP1(L107P)-mCh were transfected with wild-type REEP1-HA, REEP2-HA, REEP3-HA, or REEP4-HA and analyzed by immunostaining with αHA and αCNX antibodies. Right panels show overlays of αHA (green), L107P-mCh (magenta), and αCNX (blue). **h** Colocalization quantifications of REEP1(L107P)-mCh or REEP1wt-mCh with the ER, in cells transfected with HA-tagged REEP1-5, as analyzed in (**g**). Pearson's correlation coefficients between mCherry and αCNX are plotted with means and standard deviations. $P$-values were calculated using one-way ANOVA analysis, multiple comparisons (Tukey's method). ****$p < 0.0001$; ns, not significant. **i** Pearson's coefficients of cells as analyzed in (**g**) and quantified in (**h**), but comparing HA-tagged REEP1-5 localizations with L107P or wt REEP1-mCh. **j** Pearson's coefficients of samples as analyzed in (**g**) and quantified in (**h**), but comparing HA-tagged REEP1-5 localizations with the ER (αCNX) in cells stably expressing L107P or wt REEP1-mCh. $n$-values for (**h**–**j**), REEP1L107P-mCh/REEP1-HA, 114; REEP1L107P-mCh/REEP2-HA, 109; REEP1L107P-mCh/REEP3-HA, 104; REEP1L107P-mCh/REEP4-HA, 107; REEP1L107P-mCh/REEP5-HA, 102; REEP1wt-mCh/REEP1-HA, 115; REEP1wt-mCh/REEP2-HA, 101; REEP1wt-mCh/REEP3-HA, 107; REEP1wt-mCh/REEP4-HA, 110; REEP1wt-mCh/REEP1-HA, 113 cells. Exact $p$-values for (**f**, **h**–**j**) are listed in Source Data. All scale bars, 2 μm.

Rab3GAP1 and Rab3GAP2, which form a complex that serves as a guanine nucleotide exchange factor for Rab18 and are implicated in ER morphogenesis[50]. Another enriched protein was spartin, which plays a role in lipid droplet metabolism[51,52]. Although it is unclear how these proteins participate in REEP1 vesicle function, it is notable that mutations in Rab3GAP2 and spartin are also linked to HSP[53].

## REEP1 localization is dependent on atlastin

We wondered whether REEP1-4 proteins might cycle between the ER and the vesicular compartment. Direct budding of REEP1-4 from the ER would be consistent with our mass spectrometry results, as well as the observation that the punctae are often associated with ER tubules (Fig. 1b, c; Supplementary Fig. 1b–d; Supplementary Fig. 2g, Supplementary Video 1-4), and that REEP1 APH mutants can retain the wild-type proteins in the bulk ER (Fig. 4). Furthermore, live-cell imaging showed that REEP1-mCh occasionally localized to ER tubules or tubule tips (Supplementary Video 2, 3) or was captured as a punctum being released from the ER (Supplementary Video 4). Any recycling of REEP1-4 back to the ER would require the fusion of the vesicles with the ER and implies that the REEP1-4 proteins are packaged into the vesicles together with a membrane fusogen. A likely fusogen candidate is the membrane-bound ATL GTPase, which typically mediates homotypic ER membrane fusion[14,15]. ATLs have been reported to interact with REEP1 in co-immunoprecipitation experiments[20], and is present- although not preferentially enriched- in immunopurified REEP1wt membranes (Supplementary Fig. 8, Supplementary Data 1).

We first tested whether REEP1 and ATL proteins efficiently interact with one another, as this would facilitate co-packaging of the two proteins into budding vesicles. Streptavidin-binding peptide (SBP)-tagged REEP1 was expressed either alone or with FLAG-tagged ATL1 or ATL2 in human Expi293 cells, the membranes were solubilized in the detergent digitonin, and the samples were subjected to immuno-isolation with anti-FLAG antibodies followed by size-exclusion chromatography (Fig. 6a; Supplementary Fig. 9a–c). REEP1 indeed formed complexes with ATL1 or ATL2. Similarly, SBP-tagged REEP4 co-purified with ATL1 or ATL2 (Supplementary Fig. 9b, d).

To test whether ATL1 plays a role in REEP1 vesicle recycling, we generated stable U2OS cell lines that co-expressed a mCherry fusion of wild-type ATL1 (mCh-ATL1wt) together with mEmerald tagged REEP1 or REEP5, or GFP-tagged Sec61β. While REEP1-mEm alone resided in vesicles, co-expressing it with mCh-ATL1wt caused much of the protein to localize to the ER (Fig. 6b, Supplementary Fig. 10a). mCh-ATL1wt was found in the ER, regardless of whether it was co-expressed with REEP1-mEm, mEm-REEP5, or GFP-Sec61β, as shown by co-staining with anti-KDEL antibodies (Fig. 6b–d). Quantification of Pearson's coefficients confirmed these results (Fig. 6e). Thus, increasing the concentration of ATL moves REEP1 from vesicles to the ER. REEP1 relocalization to the ER required the TM domain of ATL1, as the co-expression of an ATL mutant lacking this domain (cytoATL1) left REEP1 in vesicles (Fig. 6e).

Next, we performed similar experiments with a fusion-defective ATL1 mutant which forms dimers that cannot undergo GTP hydrolysis[54] (mCh-ATL1 K80A). When mCh-ATL1 K80A was stably co-expressed with REEP1-mEm, both proteins colocalized to ER-independent vesicles (Fig. 6f; quantification, Fig. 6e). These ATL1 K80A- and REEP1-containing vesicles were highly dynamic, as they moved independently of ER tubules (Supplementary Video 5). mCh-ATL1 K80A-expressing cells had an intact ER network as shown with both luminal and membrane markers (Fig. 6f, Supplementary Fig. 10b) and did not show signs of ER stress (Supplementary Fig. 10c, d). When mCh-ATL1 K80A was co-expressed with mEm-REEP5 or GFP-Sec61β, all proteins remained localized to the ER, as shown by co-staining with anti-KDEL antibodies (Fig. 6g; quantification, Fig. 6e). Thus, REEP1 can move the fusion-defective ATL1 K80A mutant from the ER to vesicles. These results suggest that REEP1 and ATL are co-packaged into vesicles and require ATL-mediated fusion to return to the ER.

Finally, we tested whether the endogenous localization of a REEP1-4 protein is affected by ATLs. We utilized CRISPR to delete *ATL2* and *ATL3* (Supplementary Fig. 11a), the two predominant isoforms expressed in U2OS cells[29], and determined the localization of endogenous REEP4 with antibodies. As mentioned above, in parental cells, approximately 25% of the REEP4 punctae did not overlap with ER tubules, and a sizable pool of the punctae did not retract with the ER after nocadozole treatment (Fig. 1c; Supplementary Fig. 1c; Supplementary Fig. 11b–d). In cells lacking ATLs, the fraction of punctae with no ER overlap significantly increased (Supplementary Fig. 11d–f); expression of mCh-ATL2 wt in these cells reversed this effect (Supplementary Fig. 11d). Taken together, our results indicate that the ATLs regulate the localization of REEP1 and REEP4, promoting their cycling between the ER and a unique vesicular compartment.

## Discussion

Here we show that a subfamily of the REEPs, REEP1-4, localizes to small vesicles that do not contain established markers of known organelles. The vesicles appear to define a novel cellular compartment that has thus far escaped detection. The vesicles likely form by the ability of REEP1-4 proteins to generate high membrane curvature[21] (see model in Fig. 6h), rather than by budding of COPII-coated vesicles from ER exit sites in the normal secretory pathway. Accordingly, increasing expression of REEP1-4 proteins elevates the number of vesicles, and all mutations that compromise curvature generation localize the proteins to the bulk ER and abrogate vesicle formation. Many of the endogenous REEP1-4 molecules localize to punctae associated with the ER and may represent budding intermediates. Budding may be regulated by

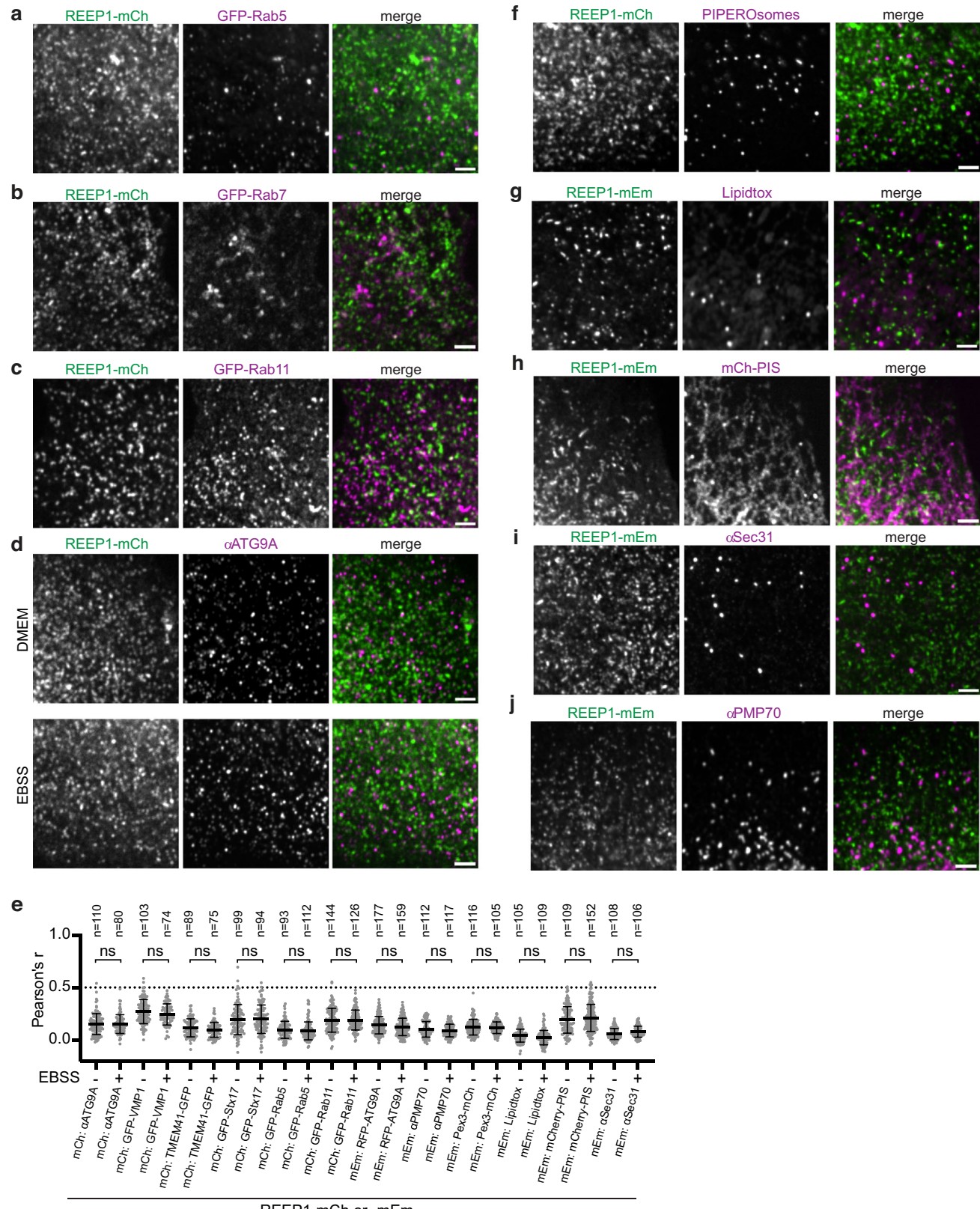

ATL, which also generates high membrane curvature[55]. Co-packaging of these proteins into the same vesicles would be facilitated by their ability to form physical complexes.

Our experiments show that ATL1 is required for the fusion of REEP1 vesicles back to the ER: When ATL1 is inactivated by the K80A mutation, both the ATL1 mutant and REEP1 remain in vesicles. These results are consistent with the established fusion mechanism of ATL; ATL molecules in vesicles would tether with those in the ER and drive fusion between the two compartments via GTP hydrolysis. After fusion with ER membranes, REEP1 might be released from ATL by being replaced with the more abundant REEP5 or reticulon proteins. Cells must maintain a balance between the concentrations of REEP1-4 on the

**Fig. 5 | REEP1 vesicles do not colocalize with known cellular compartments.**
**a** U2OS cells stably expressing REEP1-mCh were transfected with the early endosomal marker GFP-Rab5, grown in complete DMEM media, and imaged by confocal microscopy. Right panel shows the overlay between REEP1-mCh (green) and GFP-Rab5 (magenta). Scale bar, 2 μm. **b** As in (**a**), but with transfection of the late endosomal marker GFP-Rab7 (magenta). **c** As in (**a**), but with transfection of the recycling endosomal marker GFP-Rab11 (magenta). **d** As in (**a**), but analyzed by immunostaining with anti-ATG9A antibodies (αATG9A, magenta). Top row shows a cell grown in DMEM and the bottom row a cell starved in EBSS for 30 min.
**e** Pearson's correlation analysis of REEP1-mCh or -mEm colocalization with various autophagy or vesicular markers in U2OS cells cultured in full or starvation medium (− or + 30 min EBSS). See text for a description of markers. Note that all Pearson's values are under 0.5, indicating that REEP1 signals do not extensively correlate with

any marker tested. Shown are means and standard deviations. *P*-values were calculated using one-way ANOVA analysis, multiple comparisons (Sidak's method), ****$p < 0.0001$, ns, not significant. The cell number (*n*) analyzed for each condition is listed above each data set. Exact p-values are listed in the Source Data. **f** As in (**a**), but in cells coexpressing *Listeria* phospholipase C and the C1 tandem domains of protein kinase D fused to mEm, to visualize PIPEROsomes (magenta). **g** As in (**a**), but in cells stably expressing REEP1-mEm (green) and stained with the neutral lipid dye Lipidtox (magenta) to visualize lipid droplets. **h** As in (**g**), but in cells transfected with mCh-PIS (magenta), which localizes to both the ER and ER-associated punctae. **i** As in (**g**), but in cells immunostained with anti-Sec31 antibodies (αSec31, magenta) to visualize ER exit sites. **j** As in (**f**), but in cells immunostained with anti-PMP70 (αPMP70, magenta) to visualize peroxisomes.

one hand and ATL on the other, as we found that moderate overexpression of wild-type ATL1 retains REEP1 in the ER, whereas the depletion of ATLs causes a shift towards free vesicles. ATLs are normally in excess over REEP1-4 in mammalian cells[22,29,56], which limits the number of free vesicles, but boosting the expression of REEP1-4 increases the number of vesicles. In this latter situation, many of these vesicles may not contain ATL, so they may not be able to fuse back to the ER.

The generation of the vesicular compartment is physiologically important, as certain mutations in REEP1 or REEP2 that are associated with neuronal diseases abolish the formation of the vesicles and relocalize the proteins to the ER. Several of these mutations map to the APH-C of these proteins and likely reduce the proteins' ability to generate high membrane curvature. Because the mutants still form dimers, they can retain the wild-type proteins in the bulk ER. This dominant-negative effect may contribute to disease pathology. The fact that ATL1 is also commonly mutated in HSP is consistent with these proteins being involved in the same pathway. A functional interaction between REEP1 and ATL1 is supported by experiments in mice, in which the genetic deletion of *REEP1* or the expression of the ATL1 K80A mutant alone has a relatively small effect, but the combination of mutations causes profound defects in both axonal ER morphology and neurological pathologies[57].

In contrast to REEP5, a large pool of REEP1-4 molecules does not localize to the bulk ER as hitherto assumed. Thus, despite both protein families being able to generate high membrane curvature[8,21], they have distinct localizations. Our analysis shows that the N-terminal region of REEP5, which includes two amphipathic helices and the first TM segment, is a major factor determining ER localization. We speculate that the N-terminal domain allows REEP5 to generate anisotropic curvature that favors its localization to ER tubules, which have high curvature in cross-section but low curvature along the tubule length. REEP1 proteins, lacking this domain, may generate isotropic curvature that favors their localization to spherical vesicles, which have high curvature in all directions.

The exact function of the vesicles generated by REEP1-4 remains unclear. One possibility is that the budding and fusion of these vesicles allows ER tubules to shrink and grow. REEP1-4 might thus cooperate with the ATLs to generate dynamic tubules, adding another layer to how the ER tubular network remodels itself. In motor neurons, a deficiency of REEP1/2 or ATL1 may compromise the dynamics of ER tubules in axons, causing the ER to be more static or convert into nontubular structures[57], ultimately leading to HSP and HMN5B neuropathies.

REEP1 proteins may also play a role independently of ER tubule dynamics. In fission yeast, the REEP1 ortholog Rop1 is needed for the formation of phagophores, relying on its ability to generate high membrane curvature[21], and our results indicate that a subpopulation of REEP1 is recruited to phagophores in starved human cells. The association of the HSP proteins RAB3GAP2 and spartin with REEP1

membranes is also intriguing, but how these proteins would contribute to vesicle function remains to be elucidated.

## Methods

### Plasmids and antibodies
All plasmids used and generated in this study are described in Supplementary Data 2. All gblocks were purchased from IDT and Gibson assembly reactions were performed with HiFi assembly mix (NEB). Site-directed mutagenesis was performed using Pfu Turbo polymerase (Agilent). All plasmids were sequence-verified using either Sanger sequencing of the inserted/mutated region or with nanopore whole plasmid sequencing. All antibodies used in this study are listed in Supplementary Data 2.

### Mammalian tissue culture
The following human cell lines were commercially obtained: U2OS (ATCC # HTB-96), SaOS2 (ATCC # HTB-85), Bewo (ATCC # CCL-98), SKN-SH (ATCC # HTB-11), 293FT (ThermoFisher #R70007), Expi293F (ThermoFisher #A14527), and Flp-In 293 (ThermoFisher #R75007). Hela cells were a gift from B. Kingston and 293 Ampho cells were a gift from J. Brewer. All other cell lines were derived from the above lines, and all mammalian cell lines used and generated in this study are listed and described in Supplementary Data 2. Human cell lines that express high levels of REEP1, REEP2 and REEP4 were determined using the Protein Atlas database[29] (https://www.proteinatlas.org/). All adherent cells were maintained at 37 °C and 5% $CO_2$ in complete media supplemented with 10% FBS (ATCC) and penicillin-streptomycin (100 U/ml, ThermoFisher). Suspension Expi293 cells used for protein purifications were grown in Expi293 media (ThermoFisher) at 37 °C/8% $CO_2$/80% humidity in an Infors Multitron shaker.

For stable cell line generation, U2OS cells were either transfected with the indicated plasmid or infected with lentiviral or retroviral particles. Positive cells were selected using antibiotics (0.5 μg/ml G418, 0.01 μg/ml puromycin, 100 μg/ml blasticidin, 150-200 μg/ml hygromycin) to generate pooled lines. Clonal lines were isolated from pooled lines by plating single cells into 96-well plates. Lentivirus particles were packaged by co-transfection of the transfer plasmid encoding the protein of interest with psPAX2 (Addgene #12260) and pVSVG (Addgene #35616) in 293FT cells, and the media containing virus was collected ~48 h later, centrifuged to remove debris, and used directly for infection. Retrovirus particles were packaged similarly but by transfection of the transfer plasmid into 293 Phoenix Ampho cells. To generate human HEK293 cells stably expressing REEP1wt-mEm-3xHA or REEP1 L107P-mEm-3xHA for membrane immunoprecipitation experiments, linearized pcDNA5/FRT constructs containing the above coding sequences were co-transfected with pOG44 (encoding Flp recombinase) at a 1:8 (pcDNA5:pOG44) ratio with Lipofectamine 3000 into Flp-In 293 cells, and cells stably expressing the integrated constructs were selected with hygromycin (100 μg/ml). All cell lines were routinely determined to be mycoplasma-free by DAPI staining.

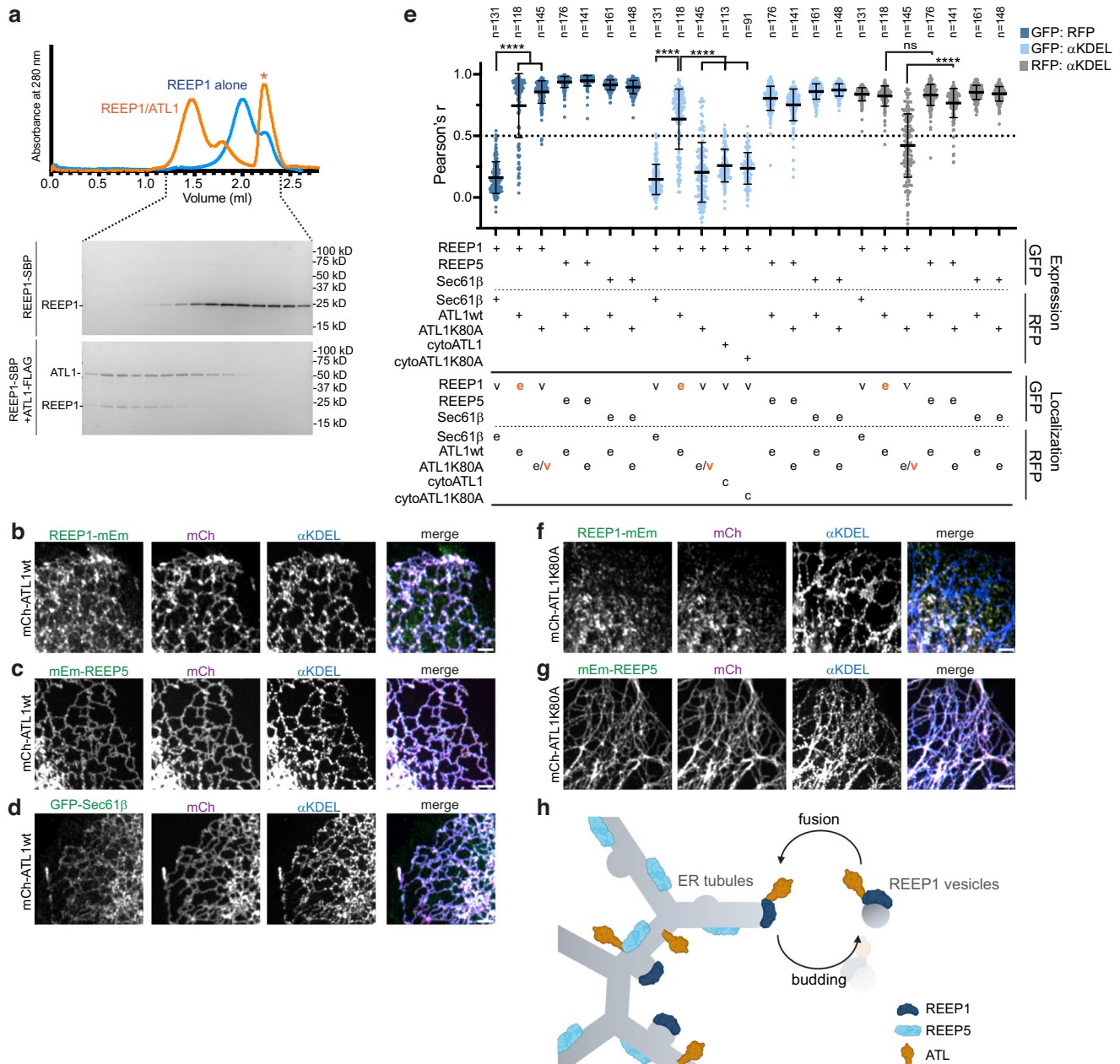

**Fig. 6 | The ATL1 fusion GTPase binds to and colocalizes with REEP1. a** SBP-tagged REEP1 (REEP1-SBP) was expressed alone or with FLAG-tagged ATL1 (ATL1-FLAG) in Expi293 cells and purified by digitonin detergent-solubilization and binding to streptavidin- (REEP1-SBP) or anti-FLAG (ATL1-FLAG/REEP1-SBP) beads. Shown is the elution profile of the samples subjected to size exclusion chromatography and the corresponding fractions as analyzed by SDS-PAGE and Coomassie blue staining. The asterisk corresponds to the 3xFLAG peptide used for ATL1-FLAG elution. **b** U2OS cells stably co-expressing REEP1-mEm and mCherry-fused wild-type ATL1 (mCh-ATL1wt) were immunostained with αKDEL antibodies and imaged using confocal microscopy. Right panel shows the overlay of REEP1-mEm (green), mCh (magenta), and αKDEL (blue). Scale bar, 2 μm. **c** As in (**b**), but with cells stably co-expressing mEm-REEP5 (green) instead of REEP1-mEm. **d** As in (**b**), but with cells stably co-expressing GFP-Sec61β (green) instead of REEP1-mEm. **e** Pearson's correlation coefficients comparing localizations of REEP1-mEm, mEm-REEP5, or GFP-Sec61β, with mCh-fusions of wild-type or mutant ATL1, all stably expressed in U2OS cells. The relative localizations of these proteins were also

compared with the αKDEL marker. As controls, cells stably co-expressing REEP1-mEm and mScarlet-Sec61β were also analyzed. The Pearson's *r* values are plotted comparing 'GFP' (mEm or GFP) to 'RFP' (mCh or mScarlet; dark blue), 'GFP' to αKDEL (light blue), and 'RFP' to αKDEL (gray) localizations. The predominant localization of each protein is indicated (e, er; v, vesicle; c, cytosolic) and changes in localization compared to controls are in orange. Shown are means and standard deviations. P-values were calculated using Kruskal–Wallis analysis, multiple comparisons test (Dunn's method). ****$p < 0.0001$; ns, not significant. The cell number (*n*) analyzed for each condition is listed above each data set. Exact *p*-values are listed in the Source Data file. **f** As in (**b**), but with cells stably co-expressing the ATL1 GTP-hydrolysis mutant K80A (mCh-ATL1 K80A, magenta). **g** As in (**c**), but with cells stably co-expressing the ATL1 GTP-hydrolysis mutant K80A (mCh-ATL1 K80A, magenta). **h** Model for REEP1 vesicle-budding from and -recycling to the ER. Created in BioRender by Y. Shibata, 2023. BioRender.com/f43f175. See text for details of the model.

U2OS *REEP4* knockout cells were generated using CRISPR and purified AsCas12a/AsCpf1. AsCas12a/AsCpf1 protein was purified[58] from an expression plasmid generated by deleting the MBP sequence from plasmid pDEST-hisMBP-AsCpf1-EC (Addgene plasmid #79007) transformed into Rosetta *(DE3)*pLysS Competent Cells (Novagen). 80 pmol of Alt-R CRISPR-Cas12a crRNA targeting sequence GGATGCTGTGTCCAGCTTATGCTT (IDT) was incubated with 63 pmol of AsCas12a/AsCpf1 protein and electroporated into $2 \times 10^5$ U2OS cells along with 39 pmol of Alt-R Cpf1 Electroporation Enhancer (IDT). Isolated clones carrying deletions in all *REEP4* gene copies were identified by PCR amplification of the targeted region with primers 5′-GTACAAGCCAAGCGAGAAAAAC and 5′-CTATTAAACA-CACCCAAACCCC and Illumina MiSeq, and further confirmed by immunoblotting with αREEP4 antibodies.

U2OS *ATL2/ATL3* knockout cells were generated similarly, using the following Alt-R crRNA target sequences: *ATL2*, CCACATCC TGGTCTTAAAGTTGCA, and *ATL3*, GGAAGGCACCAGCCACT-GAAACCA. *ATL3* was deleted from parental cells, and the *ATL2/ATL3* double KO was generated by deleting *ATL2* in *ATL3* KO cells. Knockouts were confirmed by sequencing using Illumina MiSeq with primers 5′-GCCCTCAGGTAAAACAAAATCA and 5′-AAGACAGGACATTTAAGG CGTC (*ATL2*), and 5′-CATCCTCTTGCAGGACCAC and 5′-AATGGC-CACTTTCCTTCTACAG (*ATL3*), followed by immunoblotting with αATL2 and αATL3 antibodies. All knockout sequence information is listed in Supplementary Data 2.

For transient plasmid transfections, cells were plated into 6-well plates and transfected at ~75% confluency with 0.5–1 μg of DNA using Lipofectamine 3000 (ThermoFisher). For imaging experiments, cells were trypsinized and plated onto No 1.5 cover glass (Warner Instruments) the following day and analyzed ~16-24 h later by indirect immunofluorescence. For cycloheximide-chase experiments, cells were split equivalently into two 2 cm$^2$ wells 12-16 h post-transfection and analyzed a day later.

For *REEP1* RNAi experiments, cells were reverse transfected with 20 nM of *REEP1* ON-TARGETplus *REEP1* siRNA SMARTpool siRNAs or ON-TARGETplus non-targeting siRNA oligonucleotides (Horizon # L-014235-01-0005 or #D-001810-10-05, respectively) using Lipofectamine RNAiMax (ThermoFisher). 2 days after transfection, cells were split onto coverslips and transfected again. Samples were fixed and analyzed for immunofluorescence ~24 h after the 2nd transfection.

For starvation experiments, cells were triple washed and incubated with EBSS (Millipore Sigma #E3024) for 30 min before fixation. For drug treatments, cells were incubated with nocodazole (50 μM, 1 h; Millipore #487928), puromycin (1 μg/ml, 4 or 8 h; Millipore #5.08838.0001), tunicamycin (1 μg/ml, 8 h; Millipore #T7765), thapsigargin (1.25 μM, 8 h; Millipore #T9033) or Brefeldin A (20 μM, 1 h; Millipore #203729) before fixation and immunofluorescence analysis.

For protein purification, 50 ml of Expi293 suspension cells at ~2.2 $\times 10^6$ cells/ml were transiently transfected with 50 μg of total plasmid DNA encoding human ATL1 or ATL2 tagged with a C-terminal FLAG tag, and/or human REEP1 or REEP4 tagged with a C-terminal SBP tag, using 150 μl of 1 mg/ml PEI Max (Polysciences #24765) diluted in Optimem. Expression was enhanced with the addition of 10 mM sodium butyrate (Millipore) ~ 16 h post transfection, and cells were collected 48 h later for purification.

### Indirect immunofluorescence, live imaging, and confocal fluorescence microscopy

For indirect immunofluorescence, cells were trypsinized and plated onto acid-washed No 1.5 glass coverslips (Warner Instruments) ~ 16–24 h before fixation in 4% paraformaldehyde (Electron Microscopy Services)/PBS. Cells were then permeabilized with 0.1% TritonX-100 /PBS for 5 min, blocked in 1% FBS/0.01% Triton X-100/ PBS for ~1 h, and then probed with primary antibodies. Samples were washed 3x in PBS, then incubated with secondary antibodies conjugated to Alexafluor dyes (405, 488, 568, or 647, from Thermo-Fisher). Following three more PBS washes, samples were mounted onto glass slides with Fluoromount-G (Southern Biotech). To visualize F-actin, cells were incubated with phalloidin-405 after the secondary antibody staining, washed, and mounted as above. For neutral lipid staining, cells were grown in No. 1.5 glass-bottomed dishes (Mattek #P35G-1.5-20-C), and after secondary antibody staining, Lipidtox Deep Red (ThermoFisher) was added at 1:1000 dilution in PBS and imaged directly.

For live imaging, cells were grown in glass-bottomed Mattek dishes and imaged in phenol red-free DMEM/1% FBS using an Okolab Stage Top Incubator at 37 °C and 5% $CO_2$. For Annexin V-647 staining, live cells were stained following the manufacturer's protocol (ThermoFisher) and imaged in 10 mM HEPES, 140 mM NaCl, and 2.5 mM $CaCl_2$, pH 7.4. To visualize ER membranes with DiOC6 (ThermoFisher), live cells were washed in PBS, incubated in 0.25 μM of DiOC6 diluted in PBS for 1 min, washed twice, and imaged as above.

Confocal fluorescence microscopy was performed on a Nikon Ti inverted microscope with Prior Proscan II motorized stage and shutters, a 16-bit Hamamatsu ORCA-Fusion BT sCMOS camera, Nikon LUN-F XL solid state lasers (405, 488, 561, and 640 wavelengths), and 40X, 63X, and 100X Plan Apo 1.4 or 1.44 N/A oil objectives. Images were acquired using either Metamorph (v7.8) or NIS-Elements (v5.21.03) software, and FIJI/ImageJ (v1.54i) was used for all image analyses and processing. To montage multiple fields of view, the 'Large Image' function in NIS-Elements was used with 10% edge overlap and optimal path settings. For display, fluorescence intensities of raw 16-bit depth images were linearly scaled across the entire image and then converted to 8-bit. Images showing cropped enlargements were similarly processed. Multicolored overlays were pseudocolored.

To quantify REEP protein localization with ER markers using Pearson's correlation coefficients, cells were imaged across the coverslip using a 100X Plan Apo 1.4 N/A oil objective, and all cells with a discernible reticulated ER network were analyzed. Using the 'ER' channel (i.e., FP-Sec61β, REEP1 ER-localized mutants, αCNX, or αKDEL images), ~9 × 9 μm region of interest (ROI) at the cell periphery, where the ER is a monolayer with a clearly resolved network, was manually selected and analyzed for Pearson's correlation coefficients between REEP1 and ER signals using the JaCoP or Biop JaCoP plugin. One ROI per cell was quantified and plotted in all graphs. Note that Pearson's coefficients of −1, 0, and 1 correspond to perfect anti-correlation, no correlation, and perfect positive correlation, respectively. To validate ER colocalization using this method, coefficients were calculated comparing different endogenous membrane and luminal ER markers to fluorescently tagged Sec61β, which we have previously determined to diffuse freely within ER membranes and serves as an excellent bulk ER marker[10]. From this analysis, we consider *r* values > 0.5 to represent biologically significant colocalization between two proteins localized to the ER (see Supplementary Fig 2c). Pearson's analyses of REEP protein localization with other markers were performed similarly, except ROIs were established in the cell periphery using the marker channel.

To quantify REEP punctae number /cell area and % punctae that do not overlap with the ER, ROIs were chosen similarly as above, except the relative area was increased to include more reticulated ER network when possible. Total REEP1/4 punctae were quantified as single points across the ROI using ImageJ's Find Maxima function, where the prominence was determined by setting the level to exclude anti-REEP4 signal found in *REEP4* KO cells. For REEP1 punctae counting, the prominence was determined by setting levels to exclude background signal observed in untransfected cells. Punctae number was then divided by the total ROI area × 100 μm$^2$. To calculate punctae-ER overlap, ER images were thresholded using Otsu's method and converted to a binary mask, and the number of REEP1 or REEP4 punctae that overlapped with the ER was quantified. This number was divided

by the total punctae number within the ROI, subtracted from 1, then multiplied by 100 to calculate % punctae that do not overlap with ER. One ROI per cell was graphed as a datapoint.

To compare relative REEP1 or REEP4 expression levels with punctae number, % punctae with no ER overlap, Pearson's correlation coefficients, or localization phenotypes, the mean fluorescence intensities of the αREEP1, αREEP4, or αHA signal of the whole cell were measured and subtracted of background mean intensities. To correlate REEP1-HA expression levels with different localization phenotypes, images were taken with a 63 × 1.4 N/A oil objective using identical exposure settings for the anti-HA fluorescence channel. For quantification of REEP1 and REEP4 expression levels, images were taken similarly, but with a 100X Plan Apo 1.4 or 1.44 N/A oil objective.

All quantifications were performed on raw 16-bit images, and calculations, statistical tests, and graphs were analyzed and plotted using Microsoft Excel and Graphpad Prism (v10.2.3).

## Thin-section electron microscopy to visualize APEX-DAB labeled REEP1 vesicles

U2OS cells stably expressing REEP1-APEX were fixed for 1 h in PBS containing 2% paraformaldehyde/1% glutaraldehyde, quenched with 20 mM glycine for 20 min, and then washed 5x in 100 mM sodium cacodylate buffer. Samples were then incubated with 0.5 mg/ml diaminobenzidine (Millipore #D8001)/0.1 M HCl and 10 mM $H_2O_2$ (Millipore #H1009) for 1 h on ice, and washed 5x in sodium cacodylate buffer. Samples were then fixed and stained with 1% $OsO_4$ reduced with 1.5% ferrocyanide, dehydrated sequentially in 20%, 50%, 70%, 90%, and 100% (vol/vol) ethanol on ice, and infiltrated and embedded in Epon resin. Ultrathin sections (~70 nm) were cut on a Reichert Ultracut-S microtome and mounted onto copper grids. Images were acquired on a JEOL 1200X transmission electron microscope equipped with an AMT 2k CCD camera using AMT acquisition software.

## Membrane fractionation, cycloheximide-chase, and immunoblotting

For fractionation, ~5 × 10^6 U2OS cells stably expressing REEP1-mEm were washed twice in PBS containing 1 mM phenylmethylsulfonyl fluoride (PMSF), twice with hypotonic buffer (20 mM HEPES pH 7.4, 10 mM KCl, 1 mM EGTA, 1 mM PMSF), and then collected in 1 ml of hypotonic buffer by scraping. Cells were lysed in a glass douncer, and lysates were clarified by centrifugation at 1000g for 2 min to remove unbroken cells and nuclei. KCl was then added to 150 mM, and lysates were centrifuged at 10,000g for 5 min to pellet mitochondria. 50 μl of this clarified lysate sample was collected as 'Input' and the rest divided into two tubes and fractionated by ultracentrifugation at 100,000g for 1 h at 4 °C. The supernatant containing soluble proteins was collected ('S/N'), and the pellets were washed 1x in buffer containing 0.5 M KCl to remove peripherally bound proteins to the membrane and centrifuged again at 100,000g for 30 min. One tube pellet was directly denatured with 2x Laemmli buffer for analysis ('pellet'), while the other was solubilized in buffer containing 1% dodecyl maltoside for 1 h before a final centrifugation step at 100,000g for 1 h. The resultant supernatant ('pellet +DDM') was collected and denatured with Laemmli buffer, and all fractions were analyzed by immunoblotting.

For immunoblotting, cells were lysed in modified RIPA buffer (50 mM HEPES pH 7.4, 150 mM NaCl, 1 mM $MgCl_2$, 1% TritonX-100, 0.1% deoxycholate, 0.1% SDS, with protease inhibitors), and lysates were clarified by centrifugation at ~16,000g for 10 min and denatured in Laemmli buffer.

For cycloheximide-chase analysis of REEP1-HA constructs, transiently transfected U2OS cells were treated with 50 μM cycloheximide for 6 h before lysing cells as above.

Immunoblotting was performed using standard procedures. Samples were resolved on a 4–20% gradient TGX-PAGE gel (BioRad) and transferred onto nitrocellulose. Membranes were blocked in 1–2%

milk/PBS and incubated with primary antibodies and HRP-conjugated secondary antibodies with triple wash steps after each incubation. Immunoblots were developed using western lightning ECL (Perkin Elmer) and imaged on an Amersham 800 Imager with ImageQuant TL 10.1 software (Cytiva). For some immunoblots, membranes were probed with Revert680 (LICOR) as a sample processing control and imaged on a LICOR M system using LICOR Acquisition software. All antibodies used are listed in Supplementary Data 2.

## Immunoprecipitation (IP) of REEP1wt- and L107P-mEm-HA membranes

For membrane IPs, 293 Flp-In cells expressing REEP1wt-mEm-3xHA, REEP1L107P-mEm-3xHA, or the parental control were plated onto 2 × 15 cm^2 plates the day before collection. The next day, ~4 ×10^7 total cells were washed twice in PBS containing protease inhibitors (Millipore #11873580001), twice in hypotonic buffer (20 mM HEPES pH 7.4, 10 mM NaCl + protease inhibitors), and then collected in 2 ml of hypotonic buffer by scraping. Cells were lysed by douncing in a 2 ml glass homogenizer with a tight-fitting pestle for 20 strokes, and lysates were centrifuged for 2 min at 1000g to pellet unbroken cells and nuclei. The supernatant was collected, and concentrated NaCl and EDTA were added to raise the buffer concentrations to 150 mM and 10 mM, respectively. The supernatant was then centrifuged for 10 min at 8000g to pellet mitochondria. Next, this clarified supernatant was incubated with 75 μl of anti-HA magnetic beads (Pierce; Thermofisher #88837) for ~1.5 h at 4 °C with gentle rotation. After incubation, beads were collected and washed 1x in 150 mM NaCl buffer (20 mM HEPES pH 7.4, 150 mM NaCl, 10 mM EDTA + protease inhibitors) using a DynaMag2 magnetic rack (ThermoFisher). Beads were transferred to a new tube, then washed 3x more. Samples were eluted off beads at 65 °C, 15 min, in 100 μl of 1% SDS buffer and stored at −80 °C. Six biological replicates of REEP1wt-mEm-3xHA, REEP1L107P-mEm-3xHA, or the parental control were collected for mass spectrometry analysis.

For proteomics analysis, the thawed eluates were first treated for reductive alkylation by incubation with 5 mM fresh DTT for 1 h at 37 °C, followed by addition of ammonium bicarbonate to 50 mM final concentration. Iodoacetamide was then added to 20 mM, incubated for 25 min at room temperature in the dark, and then reactions were quenched with 50 mM DTT. 100 μl of eluate was then processed with single-pot, solid-phase-enhanced sample-preparation (SP3) technology[59] to remove buffer components incompatible with mass spectrometry analysis using the following method: Eluates were incubated with 10 μl of a 1:1 mixture of hydrophilic and hydrophobic magnetic Sera-Mag carboxylate beads (50 mg/ml, washed in water; Cytiva, #24152105050250 and #44152105050250) and 110 μl of 100 % ethanol for 5 min at 24 °C at 1000 rpm in a Thermomixer shaker. Samples were then washed three times in 1 ml of 80% ethanol using a DynaMag2 magnetic rack. After the final wash, the ethanol was removed, and beads were processed for isobaric labeling and quantitative proteomics.

## Quantitative proteomics using isobaric labeling with tandem mass tags (TMT)

For protein digestion, beads were resuspended in 200 mM HEPES pH 8.5 and digested at room temperature for 13 h with Lys-C protease at a 100:1 protein-to-protease ratio. Trypsin was then added at a 100:1 ratio and the reaction was incubated 6 h at 37 °C. Peptides were separated from beads, vacuum centrifuged to near-dryness and desalted via StageTip.

For TMT labeling, a final acetonitrile concentration of ~30% (v/v) in 200 mM HEPES pH 8.5 was added along with 2 μL of TMT reagent (20 ng/μl) to the peptides in 25 μl total volume. Following incubation at room temperature for 1.5 h, the reaction was quenched with hydroxylamine to a final concentration of 0.3% (v/v) for 15 min. The TMT-labeled samples were pooled at a 1:1 ratio across all samples[60]. The

**Article**

combined sample was vacuum centrifuged to near dryness and subjected to C18 solid-phase extraction (SPE) via Sep-Pak (Waters, Milford, MA) and ready for LC-MS/MS.

To analyze the TMT-labeled samples by liquid chromatography and tandem mass spectrometry, Mass spectrometric data were collected on an Exploris480 mass spectrometer coupled to a Proxeon NanoLC-1200 UHPLC. The 100 μm capillary column was packed with 35 cm of Accucore 150 resin (2.6 μm, 150 Å; ThermoFisher) at a flow rate of 450 nl/min. The scan sequence began with an MS1 spectrum (Orbitrap analysis, resolution 60,000, 350–1350 Th, automatic gain control (AGC) target is set to "standard", maximum injection time set to "auto"). Data were acquired for 150 min per analysis. The hrMS2 stage consisted of fragmentation by higher energy collisional dissociation (HCD, normalized collision energy 32%) and analysis using the Orbitrap (AGC 300%, maximum injection time 96 ms, isolation window 0.7 Th, resolution 30,000 with TurboTMT activated). Data were acquired using the FAIMSpro interface the dispersion voltage (DV) set to 5000 V. Three data acquisitions were made, one with compensation voltages (CVs) set at −40V, −60V, and −80 V and two with three set at −30 V, −50 V, and −70 V. The TopSpeed parameter was set at 1 s per CV[61].

For data analysis, spectra were converted to mzXML via MSconvert[62]. Database searching included all entries from human UniProt reference Database (downloaded: June 2024). The database was concatenated with one composed of all protein sequences for that database in the reversed order. Searches were performed using a 50-ppm precursor ion tolerance for total protein level profiling. The product ion tolerance was set to 0.03 Da. These wide mass tolerance windows were chosen to maximize sensitivity in conjunction with Comet searches and linear discriminant analysis[63,64]. TMTpro labels on lysine residues and peptide N-termini +304.207 Da), as well as carbamidomethylation of cysteine residues (+57.021 Da) were set as static modifications, while oxidation of methionine residues (+15.995 Da) was set as a variable modification. Peptide-spectrum matches (PSMs) were adjusted to a 1% false discovery rate (FDR)[65,66]. PSM filtering was performed using a linear discriminant analysis[64] and then assembled further to a final protein-level FDR of 1%[66]. Proteins were quantified by summing reporter ion counts across all matching PSMs[67]. Reporter ion intensities were adjusted to correct for the isotopic impurities of the different TMTpro reagents according to manufacturer specifications. The signal-to-noise (S/N) measurements of peptides assigned to each protein were summed and these values were normalized so that the sum of the signal for all proteins in each channel was equivalent to account for equal protein loading. Finally, each protein abundance measurement was scaled, such that the summed signal-to-noise for that protein across all channels equals 100, thereby generating a relative abundance (RA) measurement. Keratin and trypsin contaminants were omitted from the analyzed list. P-values of the scaled signal-to-noise values from the six biological replicates were calculated by multiple unpaired, two-tailed t tests with Welch's correction using GraphPad Prism, and data were plotted using VolcaNoseR (https://huygens.science.uva.nl/VolcaNoseR/)[68].

### Protein purification of REEP-SBP/ATL-FLAG complexes
Transfected Expi293 cell pellets were solubilized in 3% digitonin in 50 mM HEPES pH 7.5, 800 mM NaCl, supplemented with DNase I and protease inhibitors, for 1.5 h at 4 °C. The lysate was centrifuged for 15 min in a TLA55 rotor (Beckman) at 112,000g to remove insoluble material. For purification of REEP1- or REEP4-SBP alone, the clarified supernatant was incubated with streptavidin agarose resin (GoldBio), eluted in buffer containing 5 mM biotin, and further purified in 20 mM HEPES pH 7.5, 150 mM NaCl, and 0.06% digitonin by size exclusion chromatography on a Superose 6 column on either an AKTA Purifier or AKTA Pure Micro system using Unicorn 7 software (Cytiva). For co-purification of ATL/REEP complexes, a similar protocol was followed

except the clarified supernatant was incubated with anti-FLAG resin (Millipore) and eluted with 0.2 mg/ml 3xFLAG peptide (Millipore). All samples were denatured in Laemmli buffer and analyzed by SDS-PAGE (4–20% TGX, Biorad) and Coomassie blue staining.

All unique biological materials generated in this study will be made available upon request.

### Statistics and reproducibility
No statistical method was used to predetermine sample size. No data were excluded from the analyses. The experiments were not randomized, and the investigators were not blinded to allocation during experiments and outcome assessment. All representative data are consistent with those obtained from two or more biological replicates unless otherwise indicated. All statistical test parameters, exact *n*-values, means, confidence levels, *p*-values, and numbers of biological replicates are summarized in the figure legends, Supplementary Information, and Source Data.

### Reporting summary
Further information on research design is available in the Nature Portfolio Reporting Summary linked to this article.

## Data availability
Mass spectrometry data have been deposited to the ProteomeXchange Consortium via the PRIDE[69] partner repository with the dataset identifier PXD055950. Raw, whole cell fluorescence images used for display are deposited in Figshare [https://doi.org/10.6084/m9.figshare.27078460], and all other raw data are available upon request. Source data are provided with this paper.

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

## Acknowledgements

The authors thank the Nikon Imaging Center/CITE and the Electron Microscopy Facility, both at Harvard Medical School (HMS), for technical assistance and use of microscopes; R. Eisert from the Protein Mass Spectrometry Facility at HMS for advice and help with initial TMT-labeling proteomics experiments; and the Taplin Mass Spectrometry Facility at HMS for MS/MS analysis of REEP4 purified complexes. We thank L. Clark for early experiments purifying REEP/ATL complexes; C. Blackstone, M. Davidson, N. Mizushima, R. Pagano, G. Voeltz, D.-H. Kim, E. Campeau, C. Vakoc, and H. Tukachinsky for plasmids; and M. Hoyer and D. Pellman for critical reading of the manuscript. S.C., L.S.A.V., and J.Z. were supported by the HMS Cell Biology Initiative for Genome Editing and Neurodegeneration. T.A.R. is a Howard Hughes Medical Institute Investigator. This article is subject to HHMI's Open Access to Publications policy. HHMI laboratory heads have previously granted a non-exclusive CC BY 4.0 license to the public and a sublicensable license to HHMI in their research articles. Pursuant to those licenses, the author-accepted manuscript of this article can be made freely available under a CC BY 4.0 license immediately upon publication.

## Author contributions

Y.S. designed this study, performed experiments, analyzed data, and wrote the paper. E.E.M. performed and analyzed imaging experiments. B.P. performed and analyzed REEP/ATL co-purification experiments. J.A.P. and S.P.G. performed the TMT-labeling and mass spectrometry analysis. S.C., L.S.A.V., and J.Z. generated the *ATL2/ATL3* and *REEP4* CRISPR-KO cell lines. T.A.R. supervised the study, performed preliminary REEP/ATL purification experiments, analyzed data, and wrote the paper.

## Competing interests

The authors declare no competing interests.
