## [Transparent Peer Review file · Nature Communications]

The membrane curvature-inducing REEP1-4 proteins generate a ER-derived vesicular compartment

Corresponding Author: Professor Tom Rapoport

Version 0:

Reviewer comments:

Reviewer #1

(Remarks to the Author)

In this study, the authors investigate the sub-cellular localization of mammalian REEP family proteins and make a surprising finding that REEP1–4 subfamily proteins localize to a novel vesicular compartment. They show that this localization requires the curvature-promoting C-terminal amphipathic helix of REEP1–4 subfamily proteins and is counteracted by ATL1. However, the physiological function of this novel compartment is unclear. The findings of this study provide useful new clues on the functions of REEP1–4 subfamily proteins.

I have the following comments and suggestions for the authors to consider.

Major points

Major point 1

The last sentence of the abstract is: "The vesicles may serve to regulate ER tubule dynamics."

In the Discussion section, the author state that:

"We propose that REEP1 proteins cooperate with ATL to generate dynamic tubule ends that can shrink and grow, adding an additional layer to how the ER tubular network remodels itself."

"REEP1 proteins would generate isotropic high membrane curvature and localize to ER tubule tips, where they cause vesicle budding and facilitate the growth and shrinkage of the tubules."

My understanding is that the authors try to assign a function to REEP1–4 subfamily proteins and propose that they act at ER tubule tips and regulate ER tubule dynamics at this location. However, there is no experimental evidence for the ER tubule tip localization of REEP1–4 subfamily proteins. There is also no evidence that perturbing the functions of REEP1–4 subfamily proteins affects ER tubule dynamics. I suggest that the authors gather additional data on these two aspects of their model.

Major point 2

Abstract

"Mutations in REEP1 proteins that compromise curvature generation, including those causing disease, relocalize the proteins to the bulk ER. These mutants interact with wild-type proteins to retain them in the ER, consistent with their autosomal-dominant disease inheritance."

Page 7

"Our results offer an explanation for the autosomal-dominant nature of APH-C disease mutations."

Page 10

"Many disease mutations map to the APH-C of these proteins, which likely reduce the ability of these proteins to generate high membrane curvature. Because the mutants still form dimers, they can retain the wild-type proteins in the bulk ER, explaining their autosomal-dominant disease phenotype."

I think these statements on the autosomal-dominant nature of the disease mutations are somewhat misleading. They may give readers a wrong impression that APH-C mutations are dominant, whereas other REEP1 disease mutations are

recessive. However, in the literature, for example Beetz et al. 2008 (PMID: 18321925), autosomal-dominant REEP1 mutations include many different types of loss-of-function mutations such as start codon loss, early frame-shift mutations, and point mutations in the loop between TM1 and TM2 (P19R and A20E). Beetz et al. 2008 explain the autosomal-dominant nature of these mutations by proposing that REEP1 is a haploinsufficiency gene. Because there is no correlation between the type of REEP1 mutations and the dominant/recessive nature of the disease, I suggest the authors do not make such prominent statements on the APH-C disease mutations being dominant or make more balanced statements on both the dominant nature of the APH-C disease mutations and the dominant nature of other REEP1 disease mutations.

Major point 3

The relationships between REEP1–4 subfamily proteins and atlastins are not fully examined. Humans have three atlastins (ATL1, ATL2, and ATL3). Do authors have any data on ATL2 and ATL3? Do ATL2 and ATL3 interact with REEP1–4 subfamily proteins and affect the localization of REEP1–4 subfamily proteins?

Minor points

Minor point 1

Page 2

"the reticulons and a branch of the REEPs (receptor expression-enhancing proteins) that in mammals includes REEP5" This branch also includes REEP6. I think REEP6 should be mentioned.

Minor point 2

Page 2

"The REEP family contains another branch which in mammals consists of REEP1-4 (called REEP1 proteins). " I think "REEP1 proteins" is not a good way of calling these proteins. Why not use "REEP1–4 subfamily proteins" or "REEP1–4 proteins", which are terms that have been used in the literature?

Minor point 3

Page 2

"most recently, to macroautophagy in fission yeast"

The other two publications (PMID: 37191320 and PMID: 37939137) reporting the autophagy role of the fission yeast ortholog should be cited.

Minor point 4

Page 3

"all point mutations that caused relocalization of REEP1-HA from vesicles to the ER are helix breakers or introduce positive charges into the hydrophobic face of the APH- C (depicted in Fig. 2g)."

S114N and A110E are not helix breakers and do not introduce positive charges.

Minor point 5

Page 5

"Disease-causing mutations are also found in the membrane-spanning region of REEP1, where the majority map to the highly conserved cytosolic loop between TM1 and TM2 (Supplementary Fig. 3a)"

"All TM mutations that cause REEP1 relocalization to the bulk ER/lipid droplets were unstable, as shown by cycloheximide-chase experiments (Supplementary Fig. 3e). "

According to Supplementary Fig. 3a, P19, A20, and S23 are residues in the loop between TM1 and TM2. Therefore, I think it is not appropriate to refer to their mutations as TM mutations. Supplementary Table 1 also has this problem.

Minor point 6

Supplementary Table 1

The first mutation listed in this table, REEP1(D113-201), is linked to the reference Beetz et al, 2008. However, I could not find this mutation in Beetz et al, 2008. It is not listed in "Table 1 Newly identified and previously reported REEP1 mutations" of Beetz et al, 2008.

Reviewer #2

(Remarks to the Author)

The paper by Shibata et al., "The membrane curvature-inducing REEP1 proteins generate a novel ER-derived vesicular compartment" uses light and electron microscopy to attempt to demonstrate that membrane curvature-generating REEP1 proteins reside in a never-before-seen vesicular compartment and identify features of the protein that determine this localization. They propose the REEP1 proteins generate the new compartment by budding from ER and then recycle back to the ER through atlastin-mediated fusion. Unfortunately, the claims made by the authors are not backed up with solid evidence, and their model of the ER budding off vesicles that then fuse back with the ER makes little sense nor has any prior support. A more likely explanation for their findings is that they result from overexpression of tagged REEP1, which is a hairpin class of proteins. Prior work has shown that when hairpin proteins are overexpressed in the ER this causes extreme thinning of ER tubules and eventual vesiculation of ER. This ER perturbation could be what the authors are observing.

Below is a list of additional comments about the paper that preclude enthusiasm for its publication.

1. The authors need to examine their “stable” cells overexpressing REEP1 to verify whether its ER morphology at the light and EM levels resemble ER morphology of control cells with no tagged REEP1 expression. This is a key control (especially using EM analysis, which needs to be volumetric EM and not a single TEM slice). Prior work has shown that even slight overexpression of hairpin proteins (of which REEP1 is one) in cells can cause significant ER morphological defects, including thin tubule formation that exclude other proteins, and eventual vesiculation of ER due to aberrant expression.
2. The authors’ main argument that overexpressed REEP1 is not perturbative is to demonstrate that REEP1 doesn’t form aggregates under their expression conditions. The real concern, however, is not this but that overexpressed REEP1, as seen for other hairpin proteins, is perturbing ER structure and inducing thin tubules (that exclude other membrane proteins) and possibly then causing ER vesiculation. This is exactly what the panels in Figure 1(b-f) suggest. In addition to examining cells with and without tagged REEP1 expression by volumetric EM, the authors need to show that tagged REEP1 or REEP4 expression does not induce UPR or calcium release from the ER, as well as other tests for the health of the ER under these conditions.
3. A further concern about their tagged REEP1 expression system is that the authors have previously shown that the yeast ortholog of REEP1 (Rop1) is associated with ER-phagy and autophagosome formation in yeast. This raises the possibility that the so-called vesicles seen by the authors are a result of ER-phagy induced by aberrant expression of REEP1. This might help explain why the APEX signal for REEP1 is many times brighter in vesicles compared to ER membranes in their EM sections. Their model of vesicles budding from ER provides no mechanism for concentration of REEP1 in these vesicles, whereas if the vesicles were part of an ER-phagy process this might be possible. If the authors want to rule out ER-phagy of REEP1, they need to do more rigorous tests than their half-hearted attempt to colocalize randomly selected autophagosome machinery.
4. The experiments with the APH-C domain truncations are greatly over-interpreted. The fact that a protein missing a domain shows ER-localization does not show that the intact protein with a different spatial distribution was NOT in the ER, nor does it imply that a putative second compartment is derived from the ER.
5. The experiments comparing REEP1 and REEP5 do not make sense. If the authors are correct that deleting residues 2-51 of REEP5 makes the protein phenomenologically similar to REEP1-4, it is then puzzling why the mutant does not colocalize at all with REEPs 1-4. The authors conclude from this that the N-terminal domain of REEP5 is important for ER tubule localization. This is strange as they argued in the paragraph before this that all REEPs are ER-targeted and induce unique budding domains that have not been identified before.
6. The authors repeatedly use punctate localization as evidence that a protein “generates” a new structure. The punctate distribution could also represent partitioning of the protein to a subdomain of ER. Lunapark and reticulons have a punctate distribution in addition to being uniform along the ER, yet the field does not talk about these proteins being in a new compartment when they localize to the puncta.
7. If the authors really think that they have discovered a new compartment, I am puzzled why they make no attempt whatsoever to suggest a) why cells would perform this energetically costly fission and fusion process, b) why no one else has described these structures despite extensive work and more than 70 years of electron microscopy, c) what proteins or functions might be associated with such a compartment. The authors claim to be able to purify the compartment by immunoprecipitation (though these data are unconvincing), so why have they not at least tried mass spec to see what proteins are present?
8. There is a fundamental flaw in the entire experimental approach arguing that lack of Pearson’s colocalization with a set of arbitrary proteins in various organelles is sufficient data to suggest that a new organelle has been discovered.
9. The model suggested by the author is effectively incompatible with the data they present in their final figure. The authors argue that budding and re-fusion of the ER-derived vesicles is mediated by atlastin and then use IPs to show similar molar ratios between atlastin and REEP1 in pulldown assays to justify this. This data alone is problematic—the authors have presented atlastin localization in figure 6 and it poorly co-localizes with the “new compartment” and instead shows mainly an ER-localization. How then can atlastin be at “equal molar ratio” in the ‘new’ compartment and not show any significant colocalization to this compartment by fluorescence microscopy?
10. Additionally, the experiments with the K80A atlastin are problematic—this mutant is well known to drive ER-stress to the point of causing prolific cell death, and the Rapaport lab has already shown that this also causes aberrant ER vesiculation in previous work.
11. The authors never show images of the whole cell, never give any measure of how representative a cell is (or comment if they are blinded when selecting cells?), and only provide the highly artifact prone measurement of the Pearson’s coefficient to quantify. The most likely explanation for all of their results, as mentioned above, is that they have generated a highly ‘unhappy’ ER by REEP1 hairpin overexpression and they are imaging aberrant and dying cells. Even in a normal coverslip, some cells will have aberrant ER vesiculation and these cells nearly always die afterwards. With this in mind, it is hard to believe the authors have discovered a “new” ER structure inside physiologically normal cells.
12. The authors end the paper with an interesting idea about REEP1-4 generating isotropic curvature but make no attempt to test this at all. The authors then go on to say they think the REEP proteins are likely required to stabilize curvature at ER tubule tips, however, the readers have just seen six figures of labeling of REEPs in which the REEPs clearly were not marking ER tubule tips.

Reviewer #3

(Remarks to the Author)

This elegant study by Shibata et al. examines the localization of the putative ER shaping REEP1 (and related family members). Unlike its ER tubule resident shaping protein cousin REEP5, REEP1 surprisingly localizes to a novel, ER derived vesicle that appears to functionally cycle between traditional ER tubules and this new vesicular compartment.

Beautiful mutational analysis by fluorescence microscopy, biochemistry, and electron microscopy confirms this new compartment, define REEP family member residence, and colocalizes associated factors like the ER fusion protein atlastin. Co-expression of mutant REEP derivatives with wild type proteins help explain the relatively complicated genetics of human mutations within the REEP family members. Overall, this study provides important and significant insight into the morphological adaptations of the ER and identifies a new compartment who function will undoubtedly provide even more important discoveries.

Minor point. The authors provide a rather exhaustive list of colocalization markers to help identify this new compartment and effectively rule out all the major organelles except one. That notable exception is the peroxisome. It would be useful to know if Pex3 or another ER derived peroxisomal membrane protein co-localized with REEP1 in this new location. This new vesicular compartment could be the elusive "pre-peroxisome" necessary for de novo reformation of peroxisomes from the ER.

Version 1:

Reviewer comments:

Reviewer #1

(Remarks to the Author)

The revision has addressed my concerns. I support the publication of the manuscript.

Reviewer #2

(Remarks to the Author)

Overall, I think the work is now publishable even though I remain skeptical about the vesicles continually fusing and budding with the ER. Publishing this work will allow others in the field to build on the interesting observations, which, although somewhat over-interpreted, are ultimately sound. The authors have identified vesicle-like structures with unknown function that are enriched in REEP1-4 and are distinct from ER. The vesicles overlap somewhat with peroxisomes, LC3 and DFCP structures (which the authors play down but is actually very interesting). Rather than asking if the REEP1-4 vesicles are possible intermediates in the formation of peroxisomes or autophagosomes (which they could readily test by knocking out REEP1-4 and looking at its effects on peroxisomes and autophagosomes in cells), the authors are content with speculating the vesicles could modulate ER structure by continually fusing and releasing from ER, providing no direct evidence for this other than showing that co-expression of REEP1 with ATL1K80A (which is inactive as a GTPase) causes both these proteins to now locate in vesicles. The figures are improved and the text comprehensible. The findings are noteworthy and will be of interest to the field. Let's now see where it takes us.

Reviewer #3

(Remarks to the Author)

The authors satisfactorily addressed my minor comments and should make a valuable addition to the Nature Communications.

Point-by-point response to Reviewers' comments for Shibata et al:

Reviewer #1 (Remarks to the Author):

In this study, the authors investigate the sub-cellular localization of mammalian REEP family proteins and make a surprising finding that REEP1–4 subfamily proteins localize to a novel vesicular compartment. They show that this localization requires the curvature-promoting C-terminal amphipathic helix of REEP1–4 subfamily proteins and is counteracted by ATL1. However, the physiological function of this novel compartment is unclear. The findings of this study provide useful new clues on the functions of REEP1–4 subfamily proteins.

I have the following comments and suggestions for the authors to consider.

Major points

Major point 1

The last sentence of the abstract is: "The vesicles may serve to regulate ER tubule dynamics."

In the Discussion section, the author state that:

"We propose that REEP1 proteins cooperate with ATL to generate dynamic tubule ends that can shrink and grow, adding an additional layer to how the ER tubular network remodels itself."

"REEP1 proteins would generate isotropic high membrane curvature and localize to ER tubule tips, where they cause vesicle budding and facilitate the growth and shrinkage of the tubules."

My understanding is that the authors try to assign a function to REEP1–4 subfamily proteins and propose that they act at ER tubule tips and regulate ER tubule dynamics at this location. However, there is no experimental evidence for the ER tubule tip localization of REEP1–4 subfamily proteins. There is also no evidence that perturbing the functions of REEP1–4 subfamily proteins affects ER tubule dynamics. I suggest that the authors gather additional data on these two aspects of their model.

We have now refrained from postulating that the REEP1-4 proteins generate tubule tips. We provide additional data showing examples of REEP1 punctae localizing to ER tubule/tubule tips (Supplementary Fig. 2g, Supplementary Video 2-3), as well as an example showing a punctum being released from the ER (Supplementary Video 4). However, it is not simple to quantify how tubule dynamics might change across the entire ER under different conditions. Thus, we have toned down the text in the Discussion, and now explicitly state that the exact function of these vesicles remains speculative (p. 12). Nevertheless, we believe the discovery of a unique vesicle population is an interesting and important finding.

Major point 2

Abstract

"Mutations in REEP1 proteins that compromise curvature generation, including those causing disease, relocalize the proteins to the bulk ER. These mutants interact with wild-type proteins to retain them in the ER, consistent with their autosomal-dominant disease inheritance."

Page 7

"Our results offer an explanation for the autosomal-dominant nature of APH-C disease mutations."

Page 10

"Many disease mutations map to the APH-C of these proteins, which likely reduce the ability of these proteins to generate high membrane curvature. Because the mutants still form dimers, they can retain the

wild-type proteins in the bulk ER, explaining their autosomal-dominant disease phenotype."

I think these statements on the autosomal-dominant nature of the disease mutations are somewhat misleading. They may give readers a wrong impression that APH-C mutations are dominant, whereas other REEP1 disease mutations are recessive. However, in the literature, for example Beetz et al. 2008 (PMID: 18321925), autosomal-dominant REEP1 mutations include many different types of loss-of-function mutations such as start codon loss, early frame-shift mutations, and point mutations in the loop between TM1 and TM2 (P19R and A20E). Beetz et al. 2008 explain the autosomal-dominant nature of these mutations by proposing that REEP1 is a haploinsufficiency gene. Because there is no correlation between the type of REEP1 mutations and the dominant/recessive nature of the disease, I suggest the authors do not make such prominent statements on the APH-C disease mutations being dominant or make more balanced statements on both the dominant nature of the APH-C disease mutations and the dominant nature of other REEP1 disease mutations.

We agree with the reviewer. We have deleted the sentence on p.7 and changed the wording on p. 10 (now p. 12, "This dominant-negative effect may contribute to disease pathology."). Although the dominant-negative effect of the disease mutations in our experiments is consistent with the autosomal-dominant nature of APH-C disease mutations, it does not exclude other disease mechanisms.

Major point 3

The relationships between REEP1–4 subfamily proteins and atlastins are not fully examined. Humans have three atlastins (ATL1, ATL2, and ATL3). Do authors have any data on ATL2 and ATL3? Do ATL2 and ATL3 interact with REEP1–4 subfamily proteins and affect the localization of REEP1–4 subfamily proteins?

We now include new data showing that REEP1-ATL2, REEP4-ATL1, and REEP4-ATL2 protein complexes can be co-purified (new Supplementary Fig. 9b-d). Additionally, we have generated an ATL2/ATL3 double knockout cell line and show that the loss of these two proteins releases more endogenous REEP4 punctae from the ER, and that expression of exogenous ATL2 can rescue this effect (new Supplementary Fig. 11). These data support our model that the ATLS generally bud together with the REEP1 proteins from the ER and affect the localization of REEP1-4 proteins.

Minor points

Minor point 1

Page 2

"the reticulons and a branch of the REEPs (receptor expression-enhancing proteins) that in mammals includes REEP5"

This branch also includes REEP6. I think REEP6 should be mentioned.

We now mention REEP6. However, we note that REEP5 is much more abundant and has been established as a tubule-shaping protein.

Minor point 2

Page 2

"The REEP family contains another branch which in mammals consists of REEP1-4 (called REEP1 proteins). "

I think "REEP1 proteins" is not a good way of calling these proteins. Why not use "REEP1–4 subfamily proteins" or "REEP1–4 proteins", which are terms that have been used in the literature?

We changed REEP1 proteins to REEP1-4 throughout the text, as requested.

Minor point 3

Page 2

"most recently, to macroautophagy in fission yeast"

The other two publications (PMID: 37191320 and PMID: 37939137) reporting the autophagy role of the fission yeast ortholog should be cited.

We have included the additional references.

Minor point 4

Page 3

"all point mutations that caused relocalization of REEP1-HA from vesicles to the ER are helix breakers or introduce positive charges into the hydrophobic face of the APH- C (depicted in Fig. 2g)."

S114N and A110E are not helix breakers and do not introduce positive charges.

We have fixed the sentence to "...introduce hydrophilic or charged residues...".

Minor point 5

Page 5

"Disease-causing mutations are also found in the membrane-spanning region of REEP1, where the majority map to the highly conserved cytosolic loop between TM1 and TM2 (Supplementary Fig. 3a)"

"All TM mutations that cause REEP1 relocalization to the bulk ER/lipid droplets were unstable, as shown by cycloheximide-chase experiments (Supplementary Fig. 3e). "

According to Supplementary Fig. 3a, P19, A20, and S23 are residues in the loop between TM1 and TM2. Therefore, I think it is not appropriate to refer to their mutations as TM mutations. Supplementary Table 1 also has this problem.

We have changed the term to "TM domain" in the text, figure legends, and in Supplementary Table 1.

Minor point 6

Supplementary Table 1

The first mutation listed in this table, REEP1(D113-201), is linked to the reference Beetz et al, 2008.

However, I could not find this mutation in Beetz et al, 2008. It is not listed in "Table 1 Newly identified and previously reported REEP1 mutations" of Beetz et al, 2008.

We have corrected the reference to Hewamadumma et al, 2009. We describe the mutant as Δ 113-201 as it is caused by a nonsense mutation that leads to a premature stop codon, resulting in a truncated protein (residue 201 is the last residue). We have added a footnote in Supplementary Table 1 for clarification and have added another footnote describing REEP2 Δ 111-252, which is caused by a similar mutation.

Reviewer #2 (Remarks to the Author):

The paper by Shibata et al., "The membrane curvature-inducing REEP1 proteins generate a novel ER-derived vesicular compartment" uses light and electron microscopy to attempt to demonstrate that membrane curvature-generating REEP1 proteins reside in a never-before-seen vesicular compartment and identify features of the protein that determine this localization. They propose the REEP1 proteins generate the new compartment by budding from ER and then recycle back to the ER through atlastin-mediated fusion. Unfortunately, the claims made by the authors are not backed up with solid evidence, and their model of the ER budding off vesicles that then fuse back with the ER makes little sense nor has any prior support. A more likely explanation for their findings is that they result from overexpression of tagged

REEP1, which is a hairpin class of proteins. Prior work has shown that when hairpin proteins are overexpressed in the ER this causes extreme thinning of ER tubules and eventual vesiculation of ER. This ER perturbation could be what the authors are observing. Below is a list of additional comments about the paper that preclude enthusiasm for its publication.

1. The authors need to examine their “stable” cells overexpressing REEP1 to verify whether its ER morphology at the light and EM levels resemble ER morphology of control cells with no tagged REEP1 expression. This is a key control (especially using EM analysis, which needs to be volumetric EM and not a single TEM slice). Prior work has shown that even slight overexpression of hairpin proteins (of which REEP1 is one) in cells can cause significant ER morphological defects, including thin tubule formation that exclude other proteins, and eventual vesiculation of ER due to aberrant expression.

We previously showed endogenous REEP1 and REEP4 punctate localization in our original manuscript, but now provide additional evidence that a sizable population of the endogenous REEP1 and REEP4 molecules in various cell lines localize to punctae that are not connected to the ER. We now start the Results section with these data (new panels Fig. 1b-c; new Supplementary Fig. 1)

The concern of the reviewer about the ER in cells stably expressing REEP1 is unjustified. The ER morphology in these cells is indistinguishable from that in the parental cells. We had shown this in the original manuscript (original Fig. 1b, c; now Fig. 1d, e), but we have now included more images of whole cells and of the cell population (new data in Supplementary Fig 2a, b, g), which demonstrate this point. The punctate structures we observe are free vesicles and not attached to the ER via thin tubules, which we show in multiple ways. For example, they move rapidly and independently of the ER in live-cell imaging (original Supplementary Video 1; new data in Supplementary Videos 2-4), they do not retract with the ER when microtubules are depolymerized (original Supplementary Fig. 4b, now Supplementary Fig. 2e; new data in Supplementary Fig. 2h), and they are not connected to the ER by protein-depleted tubules, as shown by staining with the lipophilic dye DiOC6 (new data in Supplementary Fig. 2g-h).

Thin tubules can indeed form when the reticulons or REEPs are overexpressed, as originally reported by us (Hu et al, 2008.). However, in this case, the thin tubules are filled with the overexpressed curvature-generating protein, which is clearly not the case in our current study. The vesiculation of ER tubules is not a generally observed phenomenon; it might in fact be caused by extreme overexpression. In addition, such reported ER fragments overlap with general ER markers, whereas the REEP1-4 vesicles do not.

Given the overwhelming evidence that the REEP1-4 expressing cells are normal (see also below), there is no need for volumetric EM.

2. The authors’ main argument that overexpressed REEP1 is not perturbative is to demonstrate that REEP1 doesn’t form aggregates under their expression conditions. The real concern, however, is not this but that overexpressed REEP1, as seen for other hairpin proteins, is perturbing ER structure and inducing thin tubules (that exclude other membrane proteins) and possibly then causing ER vesiculation. This is exactly what the panels in Figure 1(b-f) suggest. In addition to examining cells with and without tagged REEP1 expression by volumetric EM, the authors need to show that tagged REEP1 or REEP4 expression does not induce UPR or calcium release from the ER, as well as other tests for the health of the ER under these conditions.

We have added additional evidence that REEP1-expressing cells are neither stressed nor in other ways abnormal. We now show that the cells have an indistinguishable growth rate compared to parental cells; that they are not undergoing apoptosis by determining annexin V staining, measuring cytochrome c localization, and monitoring mitochondrial and nuclear morphology; and, that they are not undergoing ER

stress by determining the levels of BiP and looking for nuclear Xbp1 localization (new Supplementary Fig. 3). All these data confirm that the cells are healthy.

As mentioned above, we also show that the punctae are not connected to the ER by staining with the lipophilic dye DiOC6 with or without nocodazole treatment (Supplementary Fig 2g-h; Supplementary Videos 2-4). Thus, there is no reason to assume that REEP1 expression causes narrow protein-depleted tubules or ER fragmentation.

3. A further concern about their tagged REEP1 expression system is that the authors have previously shown that the yeast ortholog of REEP1 (Rop1) is associated with ER-phagy and autophagosome formation in yeast. This raises the possibility that the so-called vesicles seen by the authors are a result of ER-phagy induced by aberrant expression of REEP1. This might help explain why the APEX signal for REEP1 is many times brighter in vesicles compared to ER membranes in their EM sections. Their model of vesicles budding from ER provides no mechanism for concentration of REEP1 in these vesicles, whereas if the vesicles were part of an ER-phagy process this might be possible. If the authors want to rule out ER-phagy of REEP1, they need to do more rigorous tests than their half-hearted attempt to colocalize randomly selected autophagosome machinery.

As we stated in the original text, the bulk of REEP1 punctae are not associated with autophagosomes, with or without starvation, nor do they colocalize with major membrane proteins involved in autophagy (original Supplementary Fig. 4h, now Supplementary Fig. 7d; Fig. 5e). We now include additional panels showing localization of REEP1 with phagophore membranes marked by proteins Atg16/LC3 and Atg9A/LC3 (new Supplementary Fig. 7h, i). These results support our conclusion that only a small fraction of the REEP1 vesicles localizes to autophagosomes during starvation. The majority of vesicles remains independent of autophagy components, with or without starvation, and are clearly not generated by autophagy.

Contrary to the claim of the reviewer, our model of vesicle budding provides a simple mechanism of REEP1 concentration in budding vesicles. The high isotropic membrane curvature they generate provides a straightforward explanation for their concentration in budding vesicles and is consistent with our previous work showing that purified REEP1 proteins generate high curvature vesicles *in vitro*.

4. The experiments with the APH-C domain truncations are greatly over-interpreted. The fact that a protein missing a domain shows ER-localization does not show that the intact protein with a different spatial distribution was NOT in the ER, nor does it imply that a putative second compartment is derived from the ER.

The truncations alone do not provide evidence that the vesicular REEP1 proteins derive from the ER. However, we also present data on mutants with single residue changes in the APH that can retain the wild-type REEP1 protein in the bulk ER and show that ATLS can regulate the localization of the wild-type protein. Taken together, these results support our model that the REEP1 vesicles originate from the ER.

5. The experiments comparing REEP1 and REEP5 do not make sense. If the authors are correct that deleting residues 2-51 of REEP5 makes the protein phenomenologically similar to REEP1-4, it is then puzzling why the mutant does not colocalize at all with REEPs 1-4. The authors conclude from this that the N-terminal domain of REEP5 is important for ER tubule localization. This is strange as they argued in the paragraph before this that all REEPs are ER-targeted and induce unique budding domains that have not been identified before.

The fact that REEP5 without the N-terminal region localizes to vesicles that are distinct from those containing REEP1-4 can be explained by the lack of interaction between REEP5 and REEP1-4. This is consistent with our suggestion that these proteins bud from the ER in separate vesicles, using their independent curvature-generating abilities. Of course, all REEPs are initially localized in the ER, but only the ones that generate isotropic curvature can bud into small vesicles.

6. The authors repeatedly use punctate localization as evidence that a protein “generates” a new structure. The punctate distribution could also represent partitioning of the protein to a subdomain of ER. Lunapark and reticulons have a punctate distribution in addition to being uniform along the ER, yet the field does not talk about these proteins being in a new compartment when they localize to the puncta.

The reviewer seems to have missed that many punctae are independent of the bulk ER, as extensively shown throughout the original manuscript (original Fig. 1; Supplementary Fig. 1; Supplementary Fig. 4b, now Supplementary Fig. 2f; Supplementary Movie 1). As mentioned above, we now provide additional evidence that a large pool of both endogenous and overexpressed REEP1 proteins localize to ER-independent punctae (new Fig. 1b, c; Supplementary Fig 1c; Supplementary Fig. 2a, b, g, h; Supplementary Videos 2-4). This localization is different from the cited examples of lunapark or reticulons. In addition, the number of vesicles correlate with the expression levels of REEP1-4, which we now show in Supplementary Fig. 4i-j. Taken together with data that these proteins induce high membrane curvature *in vitro*, our results support the idea that they indeed "generate" the vesicles.

7. If the authors really think that they have discovered a new compartment, I am puzzled why they make no attempt whatsoever to suggest a) why cells would perform this energetically costly fission and fusion process, b) why no one else has described these structures despite extensive work and more than 70 years of electron microscopy, c) what proteins or functions might be associated with such a compartment. The authors claim to be able to purify the compartment by immunoprecipitation (though these data are unconvincing), so why have they not at least tried mass spec to see what proteins are present?

The exact function of the novel vesicular compartment indeed remains unknown, which we now explicitly state in the Discussion (p. 12). Arguments about the energetic costs of cellular processes have often been misleading: The reasons for ER network remodeling and dynamics are still being investigated, 40 years after its discovery. The reviewer is incorrect that 70 years of electron microscopy should have discovered this vesicular compartment. Conventional EM cannot distinguish one small vesicle from another without specific labeling methods.

Although the reviewer is mistaken that we purified these vesicles in our original manuscript, we now include new mass spectrometry data of immunoprecipitated REEP1 membranes to identify proteins that are associated with this compartment (new Supplementary Fig. 8). We identify only a handful of proteins that are enriched in REEP1 membranes, two of which are HSP disease-associated proteins (RABGAP2 and spartin). While rough ER proteins are relatively depleted from REEP1 membranes, we find that tubular ER proteins, including the atlastins, also co-precipitate (Supplementary Fig. 8; new Supplementary Table 2), suggesting a close association of these vesicles with the ER. These results are consistent with REEP1 vesicles budding from and cycling through the ER in an atlastin-dependent manner.

8. There is a fundamental flaw in the entire experimental approach arguing that lack of Pearson’s colocalization with a set of arbitrary proteins in various organelles is sufficient data to suggest that a new organelle has been discovered.

The various marker proteins were not arbitrarily chosen. To the contrary, they were selected to make the point that the novel vesicular compartment does not overlap with markers of other vesicular

compartments. All chosen proteins are established markers used by the field to identify organelles. Our mass spectrometry analysis described above also supports the novelty of this compartment.

9. The model suggested by the author is effectively incompatible with the data they present in their final figure. The authors argue that budding and re-fusion of the ER-derived vesicles is mediated by atlastin and then use IPs to show similar molar ratios between atlastin and REEP1 in pulldown assays to justify this. This data alone is problematic—the authors have presented atlastin localization in figure 6 and it poorly co-localizes with the “new compartment” and instead shows mainly an ER-localization. How then can atlastin be at “equal molar ratio” in the ‘new’ compartment and not show any significant colocalization to this compartment by fluorescence microscopy?

Our data show that the K80A mutant of ATL localizes to the vesicles. This is the basis for our model that REEP1 and ATL are packaged together into these vesicles and need ATL fusion to return to the ER. We have not been able to localize wild-type ATL to the vesicles, perhaps because its steady-state concentration is much lower in the vesicles than in the ER, and some REEP1 vesicles may indeed not contain ATL. We now note this in the text (p.12). Since it is also unclear whether ATL and REEP1 are stoichiometric in the vesicles we have also deleted this point from the text.

10. Additionally, the experiments with the K80A atlastin are problematic—this mutant is well known to drive ER-stress to the point of causing prolific cell death, and the Rapaport lab has already shown that this also causes aberrant ER vesiculation in previous work.

We are using stable cell lines that express moderate levels of ATL1 K80A. The cells divide and grow, and the ER is not fragmented as seen by the general ER marker α KDEL (Figs. 6f, g). The vesicular localization of K80A occurs only in the presence of REEP1 and is independent of the ER (Fig. 6f, g, quantification 6d). We now provide additional data showing that the ER network is intact; that, in live cells, the REEP1/K80A vesicles move rapidly and independently of the ER (Supplementary Fig. 10 a, b; Supplementary Video 5); and, that ATL1 K80A-expressing cells do not show an elevated ER stress response (Supplementary Fig. 10c, d).

11. The authors never show images of the whole cell, never give any measure of how representative a cell is (or comment if they are blinded when selecting cells?), and only provide the highly artifact prone measurement of the Pearson’s coefficient to quantify. The most likely explanation for all of their results, as mentioned above, is that they have generated a highly ‘unhappy’ ER by REEP1 hairpin overexpression and they are imaging aberrant and dying cells. Even in a normal coverslip, some cells will have aberrant ER vesiculation and these cells nearly always die afterwards. With this in mind, it is hard to believe the authors have discovered a “new” ER structure inside physiologically normal cells.

As we have discussed above, we provide more images of whole cells and data that REEP1-expressing cells are healthy and have an intact, normal ER morphology. Moreover, we present additional data of endogenous REEP1 proteins localizing to ER-independent punctae (Fig. 1b, c; Supplemental Fig. 1c), and show that the deletion of ATLs leads to a significant increase in endogenous REEP4 vesicles that are not associated with the ER (new Supplemental Fig. 11).

The images we show are representative and cells have not been cherry-picked for quantification. For quantification, cells were randomly imaged across the coverslip, and all cells with a discernable ER network were analyzed. All colocalization measurements have drawbacks (reviewed extensively by others); thus, we validated the use of Pearson’s with numerous controls comparing different ER markers to each other as well as to REEP5 (original Supplementary Fig. 1a, now Supplementary Fig. 3c).

12. The authors end the paper with an interesting idea about REEP1-4 generating isotropic curvature but

make no attempt to test this at all. The authors then go on to say they think the REEP proteins are likely required to stabilize curvature at ER tubule tips, however, the readers have just seen six figures of labeling of REEPs in which the REEPs clearly were not marking ER tubule tips.

As mentioned in the response to Reviewer #1 (point 1), we have now refrained from postulating that the REEP1-4 proteins generate tubule tips and have toned down this point in the Discussion (p. 12). We do provide additional data showing examples of REEP1 punctae localizing to ER tubule tips (Supplementary Fig. 2g, Supplementary Movie 2, 3), as well as an example of a punctum being released from the ER (Supplementary Movie 4), but we agree that tubule tip generation is speculative. Taken together, our results provide strong evidence that the REEP1-4 proteins cycle between the ER and a novel vesicular compartment, but we now make it clear in the text that the postulated effect on ER tubule dynamics is a hypothesis.

Reviewer #3 (Remarks to the Author):

This elegant study by Shibata et al. examines the localization of the putative ER shaping REEP1 (and related family members). Unlike its ER tubule resident shaping protein cousin REEP5, REEP1 surprisingly localizes to a novel, ER derived vesicle that appears to functionally cycle between traditional ER tubules and this new vesicular compartment. Beautiful mutational analysis by fluorescence microscopy, biochemistry, and electron microscopy confirms this new compartment, define REEP family member residence, and colocalizes associated factors like the ER fusion protein atlastin. Co-expression of mutant REEP derivatives with wild type proteins help explain the relatively complicated genetics of human mutations within the REEP family members. Overall, this study provides important and significant insight into the morphological adaptations of the ER and identifies a new compartment whose function will undoubtedly provide even more important discoveries.

Minor point. The authors provide a rather exhaustive list of colocalization markers to help identify this new compartment and effectively rule out all the major organelles except one. That notable exception is the peroxisome. It would be useful to know if Pex3 or another ER derived peroxisomal membrane protein co-localized with REEP1 in this new location. This new vesicular compartment could be the elusive “pre-peroxisome” necessary for de novo reformation of peroxisomes from the ER.

We have tested the colocalization of REEP1-mEm with stably expressed Pex3-mCherry as well as endogenous PMP70 (quantified in Fig. 5e and shown in new panels 5f and Supplementary Fig. 7d). REEP1 punctae do not significantly colocalize with either marker, but interestingly, in some cells they are often adjacent to Pex3-mCherry. We mention this observation in the text but have no mechanistic explanation for it.